# TRIM40 is a pathogenic driver of inflammatory bowel disease subverting intestinal barrier integrity

Sujin Kang[1], Jaekyung Kim[1], Areum Park[1], Minsoo Koh[1], Wonji Shin[1], Gayoung Park[1], Taeyun A. Lee[1], Hyung Jin Kim[1], Heonjong Han[1,2], Yongbo Kim[2], Myung Kyung Choi[1], Jae Hyung Park[3], Eunhye Lee[1], Hyun-Soo Cho[1], Hyun Woo Park[3], Jae Hee Cheon ®[4,5] ✉, Sungwook Lee ®[2] ✉ & Boyoun Park ®[1] ✉

The cortical actin cytoskeleton plays a critical role in maintaining intestinal epithelial integrity, and the loss of this architecture leads to chronic inflammation, as seen in inflammatory bowel disease (IBD). However, the exact mechanisms underlying aberrant actin remodeling in pathological states remain largely unknown. Here, we show that a subset of patients with IBD exhibits substantially higher levels of tripartite motif-containing protein 40 (*TRIM40*), a gene that is hardly detectable in healthy individuals. TRIM40 is an E3 ligase that directly targets Rho-associated coiled-coil-containing protein kinase 1 (ROCK1), an essential kinase involved in promoting cell-cell junctions, markedly decreasing the phosphorylation of key signaling factors critical for cortical actin formation and stabilization. This causes failure of the epithelial barrier function, thereby promoting a long-lived inflammatory response. A mutant TRIM40 lacking the RING, B-box, or C-terminal domains has impaired ability to accelerate ROCK1 degradation-driven cortical actin disruption. Accordingly, *Trim40*-deficient male mice are highly resistant to dextran sulfate sodium (DSS)-induced colitis. Our findings highlight that aberrant upregulation of *TRIM40*, which is epigenetically silenced under healthy conditions, drives IBD by subverting cortical actin formation and exacerbating epithelial barrier dysfunction.

The integrity of the intestinal epithelium is essential for maintaining immune homeostasis by preventing translocation of gut microbiota or harmful exogenous factors, thus disruption of this protective barrier results in IBD onset and progression[1–8]. Accumulating clinical evidence from sequencing datasets on IBD has been supporting information regarding the etiology, prognosis, and treatment of IBD[9–11]; however, the key driver of this disease has yet to be discovered. Although altered expression patterns of specific genes related to inflammation, tissue integrity, cell adhesion, and barrier function are characteristics of IBD[11–13], whether these changes are causative or consequent to IBD pathogenesis remains unclear. Additionally, the molecular basis of these changes and the stages of IBD development in which they arise are questions to be elucidated.

[1]Department of Systems Biology, College of Life Science and Biotechnology, Yonsei University, Seoul 03722, South Korea. [2]Division of Tumor Immunology, Research Institute, National Cancer Center, Goyang 10408, South Korea. [3]Department of Biochemistry, College of Life Science and Biotechnology, Yonsei University, Seoul 03722, South Korea. [4]Department of Internal Medicine and Institute of Gastroenterology, Yonsei University College of Medicine, Seoul 03722, South Korea. [5]Severance Biomedical Science Institute, Yonsei University College of Medicine, Seoul 03722, South Korea. ✉e-mail: GENIUSHEE@yuhs.ac; swlee1905@ncc.re.kr; bypark@yonsei.ac.kr

The actin cytoskeleton plays a critical role in maintaining intestinal barrier integrity; thus, epithelial dysfunction induced by cytoskeletal architecture perturbation leads to the loss of tight junctions or adherens junctions (AJs) and increases intestinal permeability[5,6,14–16]. In particular, the cellular cortex is a thin actin network bound to the plasma membrane that is present in most eukaryotic cells and forms a meshwork that prevents penetration by bacteria or viruses[17]. Actin located in the cell cortex generates cortical tension by binding to junctional molecules, and it is responsible for maintaining cellular shape, cell-cell adhesion, and junctional molecule localization, which eventually promotes intestinal homeostasis[4–6,18]. Importantly, Rho-associated coiled-coil-containing protein kinase 1 (ROCK1) is an essential kinase for actin polymerization and stabilization; therefore, it plays an important role in actin dynamics and cell migration[19–22]. Despite the functional importance of ROCK1, the specific mechanisms underlying the regulation of its expression or stability are poorly understood.

Here, we identify tripartite motif-containing protein 40 (TRIM40) as a pathogenic driver of IBD that directly targets ROCK1 for degradation, resulting in disruption of downstream signal transduction events necessary for actin cytoskeleton stabilization. This leads to failure of the epithelial barrier function and ultimately leads to IBD development. This study provides the proof of concept that paralogous *TRIM40* expression serves as a master pathological driver of chronic inflammation in the gut, and thus may be a potential biomarker or therapeutic target to limit IBD initiation and development.

## Results

### Aberrant TRIM40 upregulation links IBD and cytoskeletal regulation

To identify a potential key factor driving IBD pathogenesis, we revisited public gene expression datasets of rectum biopsy samples from patients with IBD, encompassing those with both ulcerative colitis (UC) and Crohn's disease (CD). While evaluating the highly inducible signature genes associated with IBD (fold-change [FC] > 2; $n = 527$, including genes related to cytokines, the complement system, tissue integrity, barrier function, and antimicrobial peptides), we found a significant increase in the gene expression levels of *TRIM40* and *TRIM69* in both UC and CD patients relative to healthy controls (Fig. 1a and Supplementary Fig. 1a). As altered expression of several TRIM proteins drives initiation of cancer and other immune-related diseases[23–26], we focused on both TRIM proteins as possible links to IBD pathogenesis. Unlike *TRIM69*, which exhibited diverse expression patterns in various tissues, the gene expression of *TRIM40* was barely detectable in most human tissues and human-derived cell lines, including colonic epithelial cells (HT-29, HCT116, and Caco-2), which is by contrast with TRIM31 highly expressed in intestine[27] (Fig. 1b and Supplementary Fig. 1b); moreover, its expression is much higher in IBD samples than non-IBD control intestinal tissues (Fig. 1b, red and blue squares of bottom graph). To confirm this, we analyzed three additional RNA sequencing (RNA-seq) datasets from rectum and ileum regions in IBD patients, which showed that *TRIM40* was significantly increased in samples from the rectums of patients with both UC and CD, but not in ileum tissues (Fig. 1c). Aberrant *TRIM40* overexpression was also observed in the rectal mucosa of UC patients, which this disease affects specifically[28,29] (Fig. 1d). This phenomenon was confirmed using rectal biopsy specimens from inflamed regions of patients with UC and CD (Fig. 1e and Source Data). To assess whether the protein level of TRIM40 increased in IBD patients, we performed immunohistochemistry (IHC) with a human TRIM40-specific antibody. Unlike non-IBD controls, TRIM40 protein was clearly stained in colon tissues from both UC and CD (Fig. 1f). This aberrant upregulation in the expression of the otherwise epigenetically silenced *TRIM40* occurred at a high frequency across a significant number of

patients with IBD, leading us to speculate that TRIM40 may be related to IBD pathogenesis.

Because TRIM40 protein upregulation was seen across irregular intestinal epithelium with architectural distortion and variation in the size and shape of crypts in the UC colon biopsy (Fig. 1f), we focused on the pathogenic involvement of TRIM40 in human colonic epithelial cells. To investigate this, we first explored the influence of TRIM40 overexpression on the global properties of RNA transcripts by performing RNA-seq and microarray analyses in the human colonic epithelial cell line HT-29, which stably overexpressed N-terminal Myc-tagged TRIM40 (Myc-TRIM40). Notably, the transcriptome landscape of TRIM40-overexpressing HT-29 cells showed substantial changes in genes that are signatures of IBD, including cytoskeleton-related and interferon (IFN)-stimulating genes (ISGs), which was also confirmed by quantitative PCR (qPCR) assays (Fig. 1g, h, Supplementary Fig. 1c, and Supplementary Data 1). In line with these findings, we also found that TRIM40 was significantly bound to a subset of cytoskeletal regulation-related proteins by using liquid chromatography with tandem mass spectrometry (LC-MS/MS)-based proteomics (Supplementary Fig. 1d, red-letters, and Supplementary Data 1). Given that increased ISG levels are closely correlated with actin cytoskeleton disruption[3,17,30,31], and TRIM40 overexpression appears to link the regulation of both IFNs and cytoskeleton stability, we speculated that TRIM40 upregulation might involve cytoskeleton networks and their functions. Indeed, TRIM40 mainly displayed filamentous patterns in the cytoplasm with a weak signal detected inside the nucleus (Supplementary Fig. 2a), and overexpression of green fluorescent protein (GFP)-tagged TRIM40 (GFP-TRIM40) coincided with aberrant morphological changes, including cell-to-cell repulsion, significant gaps between neighboring cells, and, eventually, dispersal (Fig. 1i, right panels). These morphological changes were not observed in control HT-29 cells, which formed tight clusters (Fig. 1i, left panels). The average edge-to-edge distance between TRIM40-overexpressing HT-29 cells was also significantly greater than that of control HT-29 cells (Fig. 1j). Similar morphological changes were observed using other colonic epithelial cells, such as Caco-2 and HCT116, as and when overexpressing TRIM40 featuring different tags fused to either the N- or C-terminus, excluding cell- or tag-specific outcomes (Supplementary Fig. 2b, c). These results indicate that high expression of epigenetically silenced *TRIM40* may serve as a potential intrinsic cue, causing fluctuations in the cytoskeletal architecture of colonic epithelial cells.

### TRIM40 subverts intestinal epithelial integrity

Because the actin cytoskeleton is central to the regulation of cell-cell adhesion[5,6,14–16], we further explored the effects of TRIM40 expression on actin dynamics. Control HT-29 cells displayed a well-organized cortical architecture that coalesced as a thin actin network layer bound to the plasma membrane (Fig. 2a, top panels). However, cells overexpressing TRIM40 exhibited cortical F-actin retraction and fragmentation, as well as irregular accumulation of F-actin at intracellular contact sites and partial TRIM40 co-localization with filamentous stretches (Fig. 2a, images #1–5 and arrows in #5), but no significant effects on monomeric G-actin expression or localization were observed (Supplementary Fig. 3a, b). This was also observed in cells expressing TRIM40 with different tags or tag positions (Supplementary Fig. 3c). Intriguingly, the altered F-actin patterns in TRIM40-expressing cells were similar to those in cells treated with cytochalasin D or phorbol 12-myristate 13-acetate, which is known to aberrantly accumulate F-actin around the nuclear membrane and disrupt cortical actin[32–34] (Supplementary Fig. 3d). These findings indicate that TRIM40 may be involved in the regulation of F-actin polymerization.

The pre-adherens junctions (AJs), which are typically observed as punctate structures aligned with cell-cell contact regions, drive membrane closure between neighboring cells by facilitating the recruitment of AJ components into AJ regions, forming cell-cell

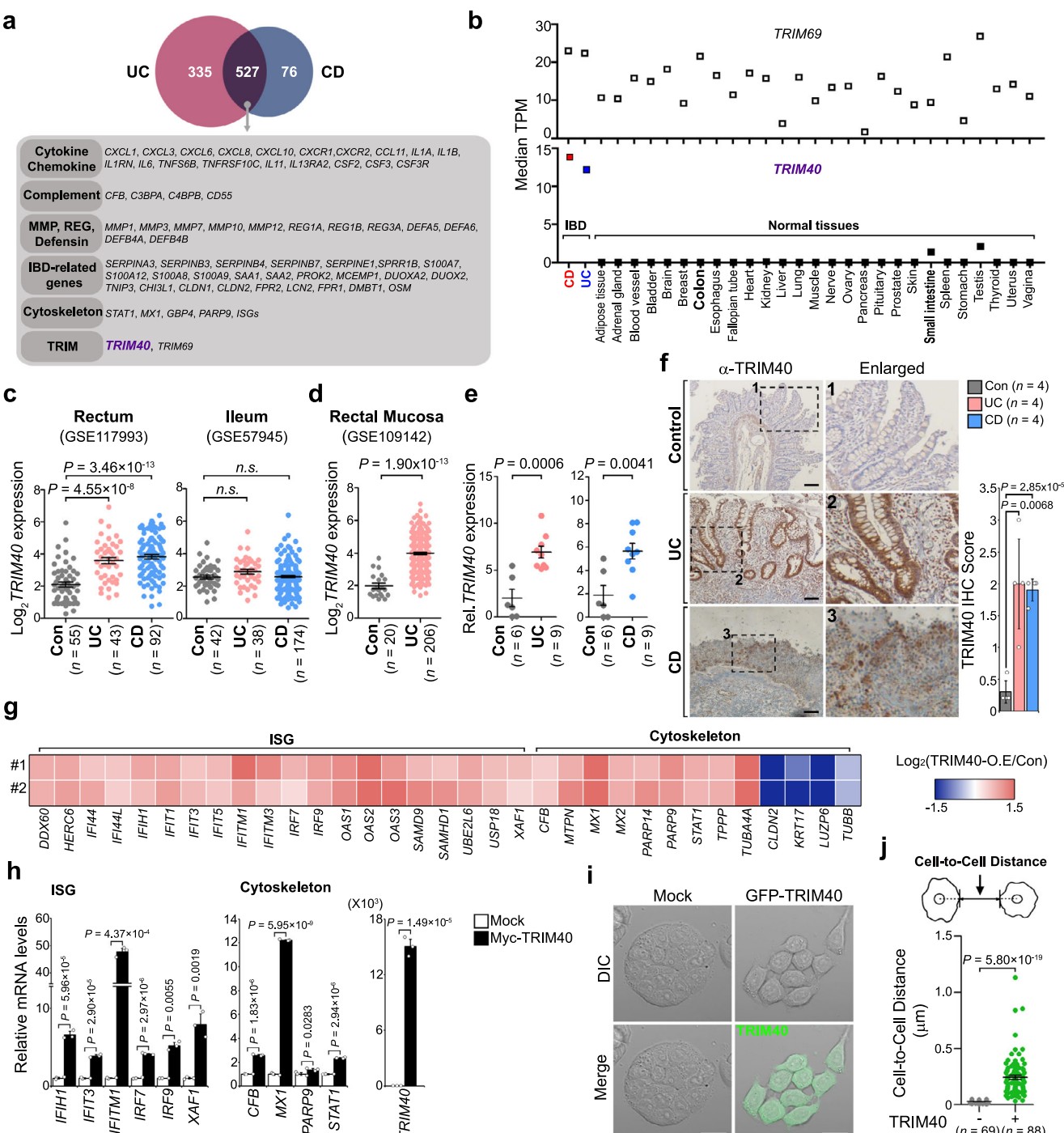

**Fig. 1 | TRIM40 upregulation is associated with IBD pathogenesis through morphological and cytoskeletal alterations. a** Venn diagram illustrating the overlap among differentially expressed genes from UC and CD patients, and healthy controls. **b** *TRIM69* or *TRIM40* expression in RNA-seq data from IBD (GSE117993) compared with normal tissues from the Genotype-Tissue Expression project. Bold letters; normal intestinal tissues. **c**, **d** Comparison of *TRIM40* expression in the rectum and ileum (**c**) or rectal mucosa (**d**) between UC, CD, and normal samples (GSE117993, GSE57945, and GSE109142). *P* values are determined by unpaired two-tailed *t* test. (*n*, numbers of patients; mean ± SD). n.s.; not significant. **e** qPCR showing *TRIM40* mRNA levels in IBD compared with normal tissues. *P* values are determined by unpaired two-tailed *t* test. (*n*, numbers of patients; mean ± SD). **f** IHC analysis of TRIM40 on control, UC, or CD patients. The mean IHC score ± SD of all samples from controls, UC, or CD was 0.3 ± 0.17, 2.0 ± 0.71, 1.9 ± 0.17, respectively (×100). *P* values are determined by unpaired two-tailed *t* test. (*n* = 4 biologically independent samples; mean ± SD). Enlarged views of the regions

(#1–3) are denoted by the black dashed squares. Scale bars, 100 μm. **g** Heat map showing the relative expression of the indicated genes by RNA-seq in control vector- or Myc-TRIM40-expressing HT-29 cells. The selected genes in the heat map were ISGs and cytoskeleton-related genes. Red or blue colors represent high- or low-fold change, respectively. O.E.; overexpression. RNA-seq results were obtained in two independent experiments (#1, #2). **h** qPCR showing relative mRNA levels of the indicated genes in control vector- or Myc-TRIM40-expressing HT-29 cells. *P* values are determined by unpaired two-tailed *t* test. (*n* = 3 biological independent experiments, mean ± SD). **i**, **j** Microscopic analysis showing morphological deformation in control or GFP-TRIM40-expressing HT-29 cells. Scale bars, 10 μm. Graphs showing comparison of cell-to-cell distance in control or GFP-TRIM40-expressing HT-29 cells. The distance was analyzed by ImageJ software (V 1.8.0) and *P* values are determined by unpaired two-tailed *t* test. (*n*, biologically independent measurements of the distance between cells; mean ± SD). Source data are provided as a Source Data file.

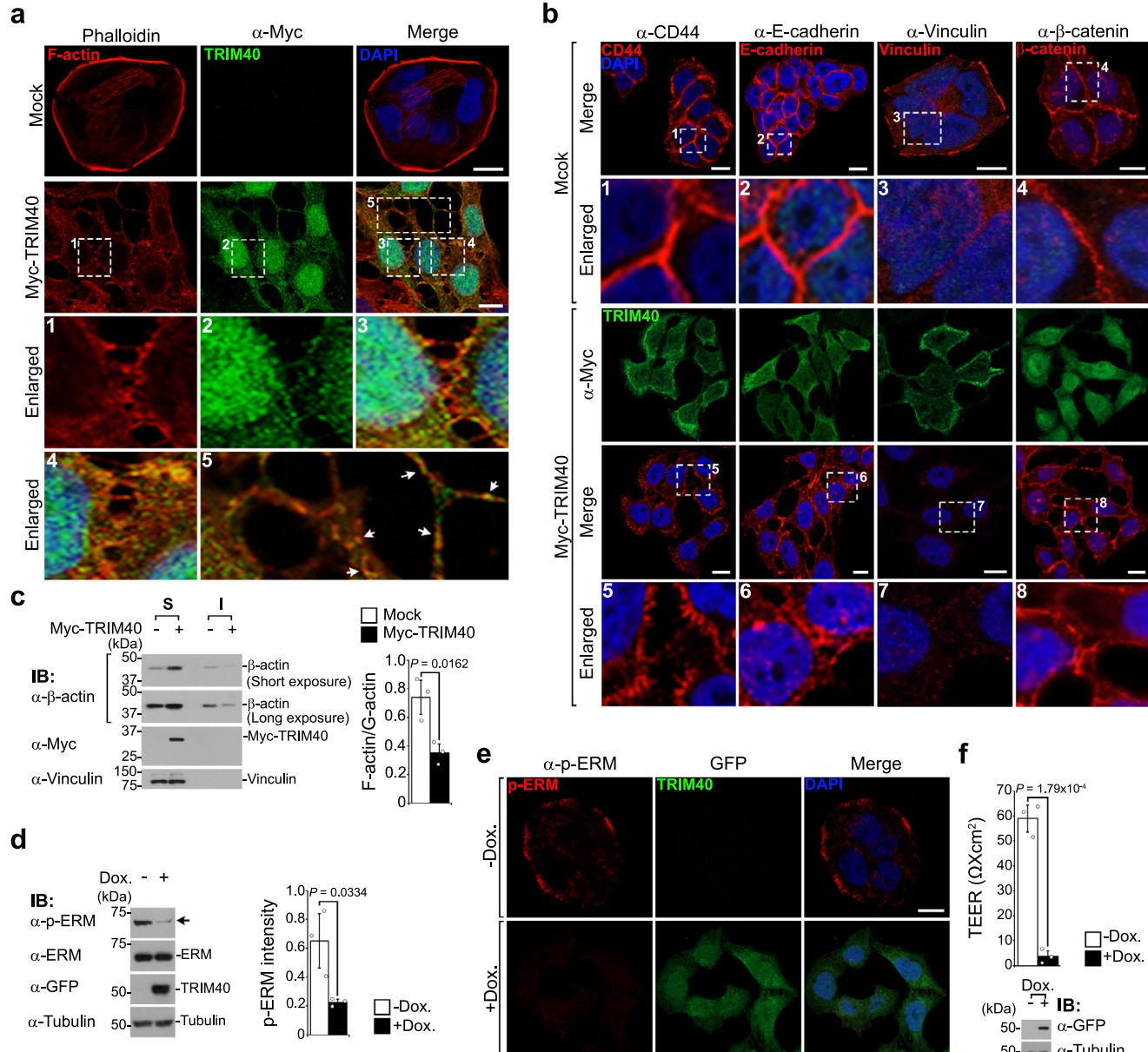

**Fig. 2 | TRIM40 disrupts epithelial integrity by destabilizing cortical actin cytoskeleton. a, b** Confocal fluorescent images of HT-29 cells expressing control vector or Myc-TRIM40. Cells were stained with anti-Myc antibody (Myc-TRIM40, green) and phalloidin (F-actin, red) (**a**). For confocal microscopy analysis of proteins located in cell-to-cell junctions, cells were stained with anti-Myc antibody (Myc-TRIM40, green) and anti-CD44, anti-E-cadherin, anti-Vinculin, or anti-β-catenin antibodies (AJ proteins, red) (**b**). Enlarged views of the regions are denoted by the white dashed squares. White arrows in image #5 represent the partial colocalization of TRIM40 with irregular accumulation of F-actin on filamentous stretches. Nuclei were stained with 4′,6-diamidino-2-phenylindole (DAPI, blue). Scale bars, 10 μm. ($n ≥ 3$ biological independent experiments). **c** Immunoblots showing β-actin levels in soluble (S) and insoluble (I) fractions from HT-29 cells expressing control vector or Myc-TRIM40. Graphs showing the ratios of F-actin to G-actin. Vinculin was used as a loading control. **d** Immunoblots showing decreased p-ERM levels in HT-29 cells temporally overexpressed GFP-TRIM40 by using a doxycycline-inducible system. Quantification of the band intensity of p-ERM (right graph). Tubulin was used as a loading control. **e** Confocal fluorescent images of GFP-TRIM40 (green) and phosphorylated ERM (p-ERM, red) in doxycycline-induced GFP-TRIM40-overexpressing cells. Cells were treated with doxycycline (2 μg ml⁻¹) for 48 h and were stained with DAPI (blue). Scale bars, 10 μm. **f** Transepithelial electrical resistance (TEER) values of Caco-2 cells temporally overexpressed GFP-TRIM40 by doxycycline treatment. *P* values are determined by unpaired two-tailed *t* test in **c, d, f**. ($n = 3$ biological independent experiments, mean ± SD). All data are representative of three independent experiments and source data are provided as a Source Data file.

adhesion zippers[14,35–38]. Notably, TRIM40-overexpressing cells failed to form adhesion zippers despite the presence of F-actin puncta at intercellular borders (Fig. 2a, images #1, #3, and #4, compared with top panels). These cells specifically contained discontinuous clusters of the major AJ components (CD44 and E-cadherin), which were located across surface membranes or at intercellular regions instead of being recruited into AJ regions (Fig. 2b, images #5 and #6, compared with #1 and #2). In particular, vinculin localizes mainly at focal adhesions in

HT-29 cells, but it translocates and accumulates at cell-cell junctions when junctional reorganization occurs in cells treated with microtubule polymerization inhibitors[39]. Similarly, we also observed that the majority (> 80-90%) of vinculin localized near focal adhesions, with small numbers of cells showing vinculin staining at cell-cell contact regions under normal conditions. Nevertheless, weak fluorescent signals of vinculin were not even detectable at cell-cell contact sites in TRIM40-expressing cells (Fig. 2b, image #7, compared with #3). This

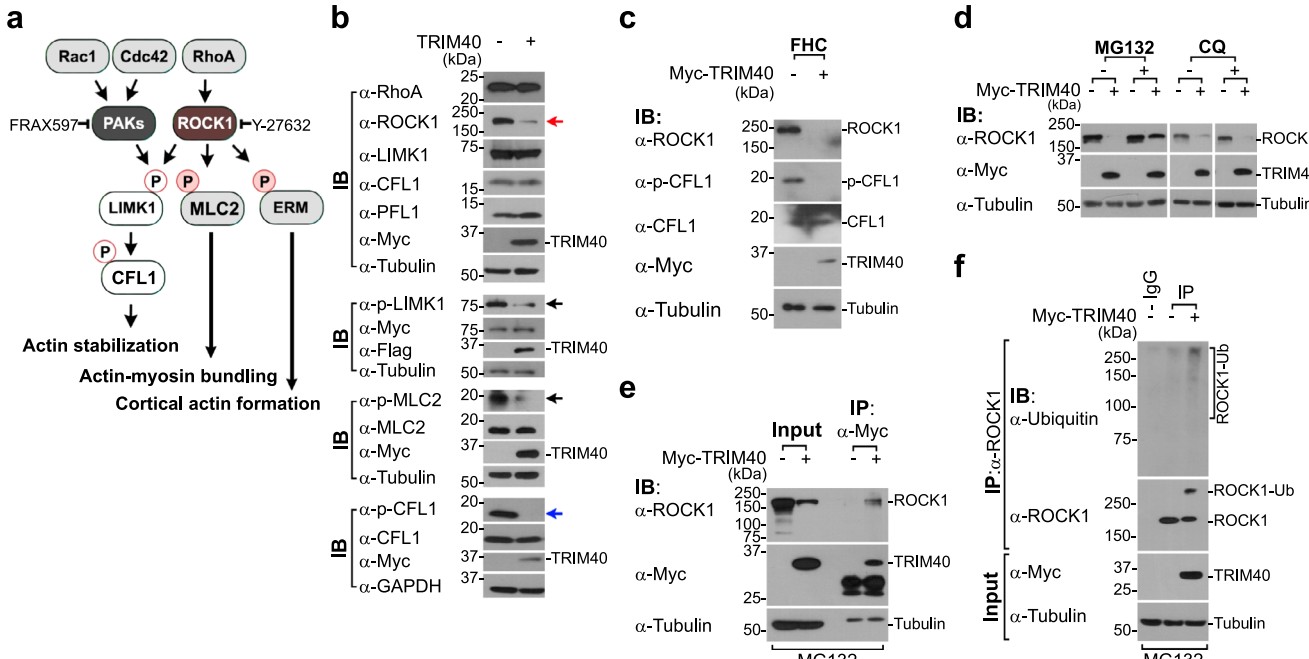

**Fig. 3 | TRIM40 regulates RhoA–ROCK1 signaling by promoting ROCK1 degradation. a** Schematic representation of actin cytoskeleton signaling pathways by Rho family GTPases. **b** Immunoblot analysis for the expression of signaling proteins associated with the actin cytoskeleton in HT-29 cells expressing control vector or Myc-TRIM40. Cell extracts were analyzed using the indicated antibodies. For p-LIMK1 levels, lysates from cells expressing LIMK1-Myc with control vector or Flag-TRIM40 were immunoblotted with the indicated antibodies. Tubulin and GAPDH were used as a loading control. **c** Immunoblot analysis of ROCK1, p-CFL1, and CFL1 levels in FHC cells expressing control vector or Myc-TRIM40. Cell lysates were analyzed using the indicated antibodies. Tubulin was used as a loading control. **d** Immunoblot analysis of ROCK1 protein levels in HT-29 cells expressing control vector or Myc-TRIM40 after treatment with MG132 (10 μM) or chloroquine

(CQ, 10 μM) for 12 h. Cell lysates were analyzed using the indicated antibodies. Tubulin was used as a loading control. **e** Co-immunoprecipitation of TRIM40 with ROCK1 in HT-29 cells expressing control vector or Myc-TRIM40. Cells were treated with MG132 (20 μM) for 4 h and cell lysates were immunoprecipitated with anti-Myc antibodies and then analyzed by immunoblotting with the indicated antibodies. **f** Immunoblots showing ROCK1 ubiquitination in HT-29 cells expressing control vector or Myc-TRIM40. Cells were treated with MG132 (10 μM) for 12 h and lysed. Cell lysates were immunoprecipitated with anti-ROCK1 antibodies and analyzed by immunoblotting with indicated antibodies. Tubulin was used as a loading control. All data are representative of three independent experiments and source data are provided as a Source Data file.

distribution of AJ components and the loss of interdigitation between neighboring cells differed from these features in control cells, which featured densely-packed, continuous AJ regions along the vast majority of lateral cell borders, without voids (Fig. 2b, images #5–7, compared with #1-3). As irregular E-cadherin puncta with large numbers of voids result in junctional defects on lateral membranes[40], our findings indicate that TRIM40-driven discontinuous AJ clusters cause perturbations in cell-cell adhesion. These perturbations were not attributable to effects on AJ expression levels, as no differences were observed in total protein or cell surface protein levels between control and TRIM40-overexpressing cells (Supplementary Fig. 3e, f). Moreover, similar puncta patterns were observed for cadherin-attached β-catenins in TRIM40-overexpressing cells (Fig. 2b, image #8, compared with #4), resulting in a significant increase in transcriptional enhanced associate domain (TEAD) activation by TRIM40 expression levels without altering β-catenin protein expression levels (Supplementary Fig. 3g, h). As the β-catenin-nuclear Yes-associated protein (YAP)-TEAD activation has been reported to be associated with intracellular F-actin destabilization[41], our findings indicate that TRIM40-driven aberrant β-catenin accumulation at intercellular borders is correlated with the negative regulation of F-actin stabilization.

To verify the possibility that TRIM40 contributes to F-actin disruption, enriched G- and F-actin fractions were isolated from control and TRIM40-overexpressing cells. Compared with control cells, the amount of soluble G-actin in TRIM40-overexpressing cells increased significantly; notably, F-actin decreased by a similar amount (Fig. 2c). Because the phosphorylation of the ezrin, radixin, and moesin protein family (ERM) is an essential outcome of cellular upstream processes

that activate to stably polymerize cortical F-actin[19,21,22], we examined the effects of TRIM40 overexpression on phosphorylated ERM (p-ERM) levels. Importantly, TRIM40 overexpression decreased ERM phosphorylation without effects on endogenous ERM protein levels, which was consistent with results from confocal microscopy analysis (Supplementary Fig. 4a, b). This decrease in p-ERM levels eventually disrupted epithelial membrane integrity through the near-complete inhibition of electrical resistance in epithelial cells (Supplementary Fig. 4c). To confirm these results, we generated a Tet-On system for doxycycline-inducible TRIM40 expression, which allowed us to overexpress TRIM40 in a controlled manner and accurately monitor the specific effects of TRIM40 overexpression in epithelial cells. Consistently, we observed that p-ERM levels decreased significantly in HT-29 cells after doxycycline treatment (Fig. 2d, e), while the expression of TRIM40 was maximized (Supplementary Fig. 4d). Moreover, epithelial membrane integrity was also disrupted in cells that overexpressed TRIM40, as indicated by measurement showing electrical resistance inhibition of >80% in the epithelial cells (Fig. 2f). Collectively, our findings suggest that pathological function gain due to overexpression of the typically epigenetically repressed *TRIM40* is associated with the perturbation of epithelial cell integrity caused by the disruption of a series of molecular processes, including ERM phosphorylation, underlying cortical F-actin formation and stabilization.

## TRIM40 is an E3 ligase directly targeting ROCK1
The most prominent Rho family GTPases-Rac1, Cdc42, and RhoA- act as molecular switches to activate two major kinases, p21-activated kinase (PAK) and Rho-associated coiled-coil-containing protein kinase

1 (ROCK1)[19–22] (Fig. 3a). Activation of these kinases triggers the phosphorylation of key downstream effectors, including Lim kinase 1 (LIMK1), myosin light chain 2 (MLC2), and ERM, which eventually facilitates F-actin formation, stabilization, and cortical actin formation[21,22]. Inhibition of ROCK1 with Y-27632 resulted in disruption of cortical F-actin formation, similar to the morphological changes observed in cells with TRIM40 overexpression, whereas PAK inhibition using FRAX597 did not show similar results (Supplementary Fig. 5a, images #1 and #2). Together with these results, the fact that TRIM40 drives inhibition of ERM phosphorylation promoted us to perform a broad screening of protein expression associated with ROCK1 and its downstream signaling cascade. In TRIM40-overexpressing HT-29 epithelial cells, ROCK1 protein levels but not mRNA levels were remarkably decreased (Fig. 3b, red arrow, and Supplementary Fig. 5b); however, similar effects were not observed for proteins either upstream or downstream of ROCK1, including RhoA, LIMK1, MLC2, cofilin1 (CFL1), and profilin1 (PFL1) (Fig. 3b). We also observed that PAK1 protein levels were not affected by TRIM40 overexpression, as no differences were observed in FRAX597-treated cells (Supplementary Fig. 5c). The effects of TRIM40 overexpression on ROCK1 protein expression were also detectable in other colonic epithelial cell lines (HCT116 and Caco-2), and no tag-specific effects were detected (Supplementary Fig. 5d, e). Accordingly, TRIM40 overexpression blocked the phosphorylation of both LIMK1 and MLC2, which are known substrates of ROCK1, in addition to the observed inhibitory effects on ERM phosphorylation (Fig. 3b, black arrows). Ultimately, inhibition of LIMK1 phosphorylation resulted in the disruption of CFL1 phosphorylation (Fig. 3b, blue arrow). This was confirmed using the normal colonic epithelial cell line FHC, showing that TRIM40 overexpression clearly degrades ROCK1 and subsequently disrupts CFL1 phosphorylation in FHC cells (Fig. 3c). Moreover, phosphorylated MLC2 failed to co-localize with cortical F-actin in TRIM40-overexpressing HT-29 cells, consistent with the immunoblot results showing the failure of MLC2 phosphorylation (Supplementary Fig. 5f). In line with previous reports showing that disruption of cell-cell contact tends to increase RhoA activity[42,43], we also observed a significant increase in RhoA activity in TRIM40-expressing cells despite similar RhoA protein levels in wild-type TRIM40 (Supplementary Fig. 5g). Our findings suggest that TRIM40 selectively downregulates ROCK1, preventing a series of phosphorylation events that are critical for facilitating F-actin formation and stabilization, resulting in the disruption of epithelial cell integrity.

Because TRIM40 has an RBCC (RING, B-box, and coiled-coil) ubiquitin ligase domain, we speculated that TRIM40 directly targets ROCK1 for degradation. Notably, proteasome inhibition with MG132 prevented the TRIM40-mediated reduction in ROCK1 protein level, whereas lysosome inhibition with chloroquine had no effect (Fig. 3d). Moreover, following MG132 treatment, TRIM40-expressing cells showed the physical interaction of TRIM40 with ROCK1 and considerable accumulation of ROCK1 ubiquitin conjugates (Fig. 3e, f). Additionally, the result from surface plasmon resonance confirmed the direct binding of TRIM40 with ROCK1 ($K_d = 2.25$ μM, Supplementary Fig. 5h). To further ascertain which TRIM40 functional domain is responsible for ROCK1 degradation activity, we generated TRIM40 mutants lacking the RING (TRIM40ΔRING), B-box (TRIM40ΔBB), coiled-coil (TRIM40ΔCC), or C-terminal region (TRIM40ΔCT) (Fig. 4a).

TRIM40ΔCC protein was barely detectable, despite similar mRNA expression levels as the wild-type TRIM40, whereas other mutants were sufficiently stable, indicating that the coiled-coil domain may be involved in TRIM40 protein stability (Supplementary Fig. 6). Notably, TRIM40ΔRING failed to induce ROCK1 ubiquitination (Fig. 4b), resulting in partial restoration of ROCK1 protein levels (Fig. 4c, bottom graph). This failure of TRIM40 to degrade ROCK1 was also confirmed by using an E3 ligase-dead TRIM40 mutant with serine in place of cysteine 29 in the RING domain (TRIM40-C29S), which lost its

functional activity (Fig. 4d). Confocal analysis also showed a similar partial restoration of cortical actin in TRIM40-C29S-expressing cells, as observed in cells expressing TRIM40ΔRING (Fig. 4e). This finding indicates that the E3 ligase activity of TRIM40 is necessary, albeit insufficient, to direct ROCK1 to the ubiquitin-proteasome pathway for degradation. In addition, TRIM40ΔBB fully lost its binding affinity for ROCK1 (Fig. 4f; lane #4, compared with lane #2), leading to the loss of both TRIM40-driven ROCK1 degradation and cortical actin disruption (Fig. 4c, e). Particularly, the creation of a dimeric form of TRIM40ΔCT led us to speculate that the C-terminal region structurally prevents TRIM40 dimerization (asterisks in Fig. 4c, f), which may be a determining factor affecting its ability to destroy epithelial integrity. Indeed, TRIM40ΔCT had no effect on ROCK1 degradation or interaction (Fig. 4c and lane #5 in f) and eventually lost its cortical actin disruption functionality (Fig. 4e). Moreover, neither TRIM40ΔBB nor TRIM40ΔCT exhibited negative effects on cell-cell distance, and the electrical resistance of cells was only partially affected by TRIM40ΔRING (Fig. 4g, h), indicating that the functional abilities of the TRIM40 mutants to affect ROCK1 degradation were directly reflected in both cortical actin formation and cell-cell contacts. Taken together, these results demonstrate that TRIM40 acts as an E3 ligase directly targeting ROCK1 through physical interaction with ROCK1 via the B-box and C-terminal region, and the facilitating degradation of ROCK1 results in the subversion of epithelial cell integrity.

### Trim40-deficient male mice are resistant to DSS-induced colitis

To explore the physiological effects of TRIM40 on F-actin destabilization and cortical actin disruption in vivo, we first examined the expression levels of *Trim40* in various mouse tissues. Similar to human gene expression patterns, mouse *Trim40* was silenced across most mouse tissues, but weakly expressed only in the cecum, intestine, and colon samples, albeit at a very low level, compared with the expression of the intestinal *Trim31*[27] (Supplementary Fig. 7a). The dextran sulfate sodium (DSS)-induced murine colitis model is often used when studying UC due to many similarities between the pathology displayed in this model and human UC pathology[44]. However, the mechanisms through which DSS induces persistent chronic inflammation remain unclear. Intriguingly, DSS-treated mice showed a significant increase in *Trim40* expression, consistent with results from analyses of public RNA-seq data from the rectum of DSS-treated mice (Fig. 5a and Supplementary Fig. 7b). To confirm this, we examined the *Trim40* over-expression in colon tissues from DSS-induced mice by performing an RNA in situ hybridization assay with a labeled RNA probe specifically recognized mouse *Trim40* mRNA. We observed significantly increased *Trim40* expression in the rectum and colon tissues of DSS-treated mice (Fig. 5b and Supplementary Fig. 7c). Moreover, *Trim40* expression was upregulated in the distal colon of mice that received intraperitoneal administration of lipopolysaccharide (LPS), but little effect was observed on *Trim31* expression (Supplementary Fig. 7d), which was consistent with results from public RNA-seq data with intestinal epithelial cells stimulated by LPS (Supplementary Fig. 7e). These results indicate that DSS- or LPS-induced inflammation can stimulate *Trim40* expression. In turn, these stimuli, such as infections, could possibly lead to transcriptional initiation of silenced *Trim40* gene and the loss of epithelial cell integrity, and subsequently acceleration of chronic inflammation.

This finding led us to hypothesize that *Trim40*-deficient mice might be resistant to DSS-induced colitis due to the inability to stimulate *Trim40* expression. To test this hypothesis, we generated *Trim40*-deficient (*Trim40⁻/⁻*) mice and examined the pathological effects of *Trim40* loss on murine colitis (Supplementary Fig. 8a, b). Under physiological conditions, histological architectures of *Trim40⁻/⁻* mice, including cell numbers and epithelial architecture, were similar to that of wild-type (WT) mice (Supplementary Fig. 8c). Following DSS administration, *Trim40⁻/⁻* mice developed clinical symptoms of UC,

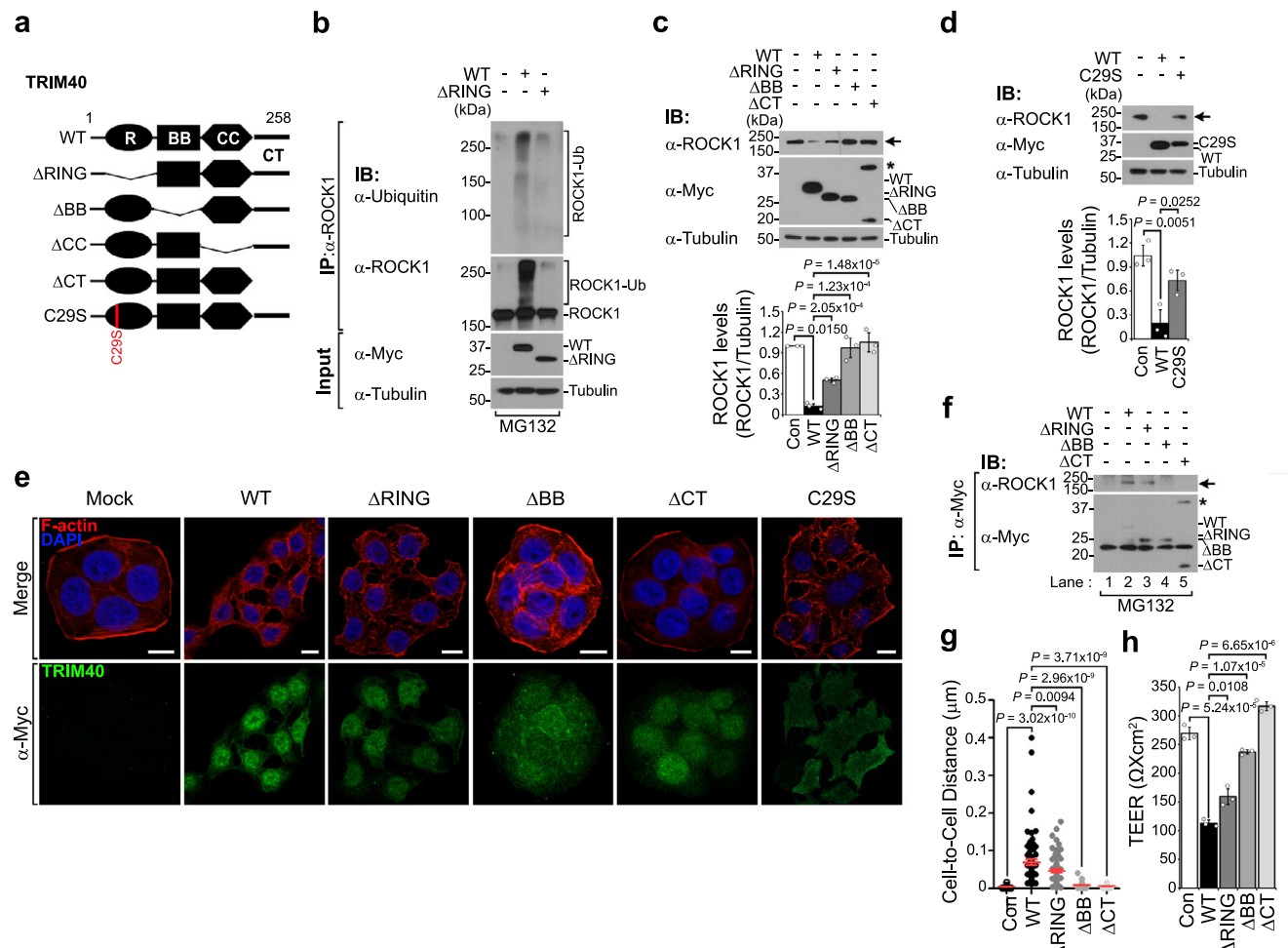

**Fig. 4 | The RING, B box, and C-terminal domains of TRIM40 are required for ROCK1 degradation. a** Schematic representation of full-length TRIM40 (WT) and TRIM40 mutants. ΔRING, ΔBB, ΔCC, or ΔCT lacks RING, B-box, coiled-coil, or C-terminal region, respectively. TRIM40-C29S substitutes cysteine with serine at 29 residue. **b** Immunoblots showing ROCK1 ubiquitination in control vector-, WT-TRIM40-, or TRIM40ΔRING-expressing HT-29 cells. **c** Immunoblots of ROCK1 levels in HT-29 cells expressing WT-TRIM40 or TRIM40 mutants. Quantification of ROCK1 band intensity relative to tubulin band intensity (bottom graph). The asterisk, dimeric forms of TRIM40ΔCT. **d** Immunoblot analysis of ROCK1 levels in HT-29 cells expressing control vector, WT-TRIM40, or TRIM40-C29S. Quantification of ROCK1 band intensity relative to tubulin band intensity in lysates (bottom graph). **e** Confocal images showing cortical F-actin in HT-29 cells expressing control vector, WT-TRIM40, TRIM40 deletion, or TRIM40-C29S. Cells were stained with phalloidin for cortical F-actin (red) and anti-Myc antibody for WT-TRIM40, TRIM40 deletion, or TRIM40-C29S mutants (green). Nuclei were stained with DAPI (blue). Scale bars,

10 μm. **f** Co-immunoprecipitation of TRIM40 mutants with ROCK1 in HT-29 cells expressing control vector, WT-TRIM40, or TRIM40 deletion mutants. After MG132 (20 μM) treatment, lysates were immunoprecipitated with anti-Myc antibody, and analyzed by immunoblotting with the indicated antibodies. *; dimeric forms of TRIM40ΔCT. **g** Graph showing comparison of cell-to-cell distance in HT-29 cells expressing control vector ($n = 52$), WT-TRIM40 ($n = 77$), TRIM40ΔRING ($n = 83$), TRIM40ΔBB box ($n = 51$), and TRIM40ΔCT ($n = 47$). The distance was analyzed by ImageJ software (V 1.8.0) and $P$ values are determined by unpaired two-tailed $t$ test. ($n$, biologically independent measurements of the distance between cells; mean ± SD). **h** TEER assay for intestinal epithelial barrier function in Caco-2 cells expressing control vector, WT-TRIM40, or TRIM40 deletion mutants. $P$ values are determined by unpaired two-tailed $t$ test in (**c**, **d**, **h**). ($n = 3$ biological independent experiments, mean ± SD). All Data are representative of three independent experiments and source data are provided as a Source Data file.

such as loss of body weight and shortening of colon length, but these symptoms were not as severe as those observed in control mice (Fig. 5c, d). Particularly, although body weight decreased in $Trim40^{-/-}$ mice, it gradually recovered after DSS removal, which was not observed in control mice (Fig. 5c). Due to the severity of UC symptoms, most control mice died within 10 days of DSS administration, whereas approximately 70% of $Trim40^{-/-}$ mice survived for over 2 weeks (Fig. 5e). These phenomena were also observed in a 2,4,6-trinitrobenzene sulfonic acid (TNBS)-induced mouse colitis model, used to develop mucosal inflammation-induced colitis[45,46] (Supplementary Fig. 8d, e, g). Histopathology also confirmed that epithelial barrier, crypt and goblet cell architectures remained intact in $Trim40^{-/-}$ mice, even after DSS administration (Fig. 5f, images #2 and #4). By contrast, control mice showed severe epithelial barrier breakdown, with fewer

goblet cells and crypt deformation (Fig. 5f, images #1 and #3). Furthermore, no visible lymphocyte infiltration was observed in the lamina propria or epithelium of DSS-treated $Trim40^{-/-}$ mice (Fig. 5f, image #2, compared with #1). These extensive pathological patterns were clearly observed in both the rectum and distal colon of DSS-treated control mice but not in the proximal and mid colon, because DSS-induced pathology is specific to the distal colon and rectum, where an enormous number of bacteria live[44] (Fig. 5f and Supplementary Fig. 8f). Moreover, DSS-induced acute gene expression of pro-inflammatory cytokines and chemokines, but not anti-inflammatory $Il10$, was substantially attenuated in $Trim40^{-/-}$ mice (Fig. 5g). This was confirmed via TNBS-induced colitis, which showed that TNBS-induced acute $Cxcl1$ expression was considerably attenuated in $Trim40^{-/-}$ mice (Supplementary Fig. 8g). Consistent with in vitro results in human

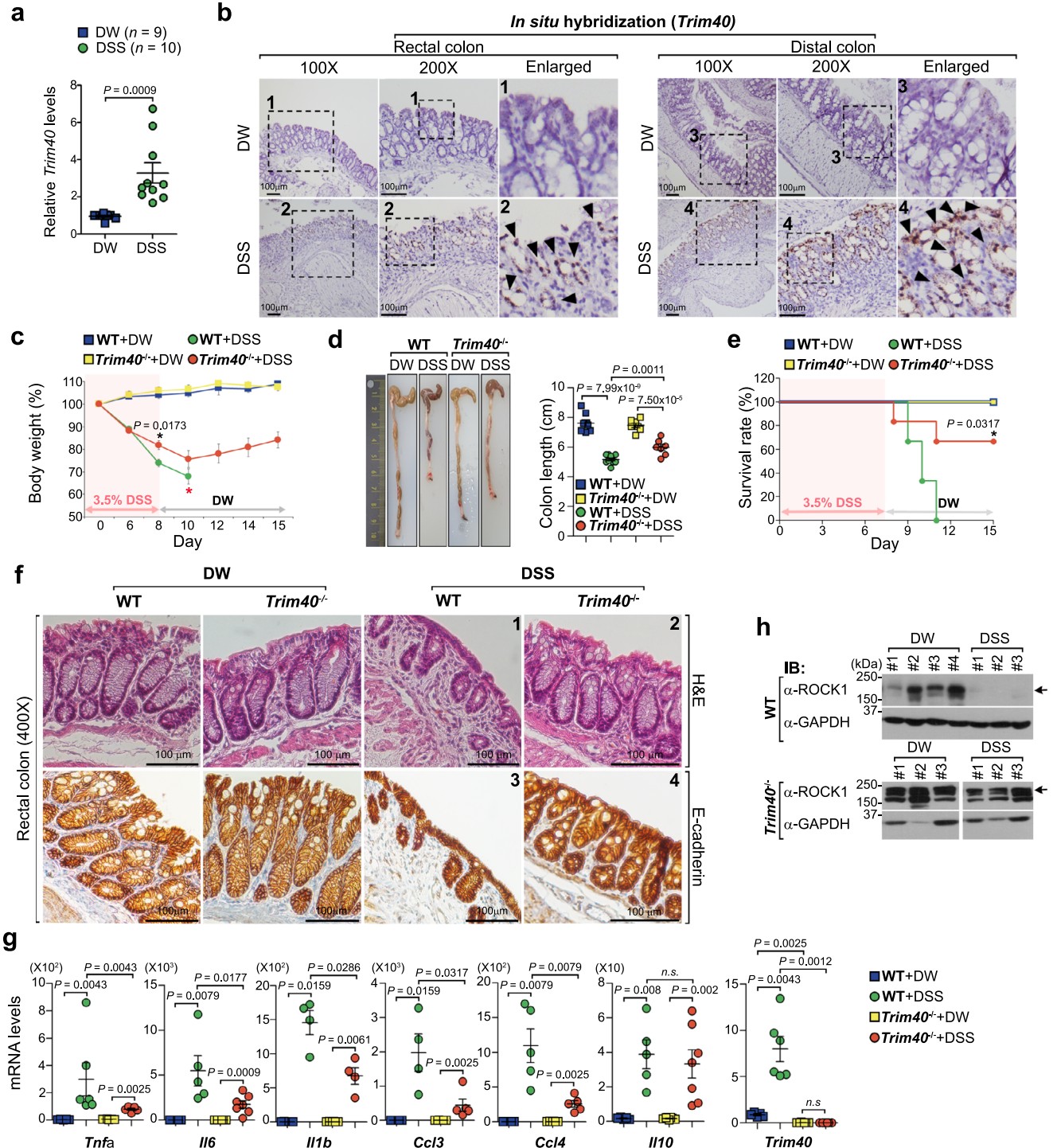

**Fig. 5 | Trim40-deficient mice were less susceptible to DSS-induced colitis.**
**a** qPCR showing relative mRNA levels of *Trim40* in 2% DSS-induced colitis. *Trim40* expression was analyzed in colon tissue from WT male mice (DW, *n* = 9; DSS, *n* = 10). *P* values are determined by unpaired two-tailed *t* test. (*n*, numbers of mice; mean ± SD). **b** ISH showing *Trim40* mRNA levels in rectal or distal colon tissues from WT male mice with 2% DSS. Brown (black arrowheads) indicates *Trim40* mRNA expression. Enlarged views of the regions are denoted by the black dashed squares (#1–4). Scale bars, 100 μm. **c** Relative loss of body weight in male mice with 3.5% DSS. WT (DW, *n* = 3; DSS, *n* = 6), *Trim40⁻/⁻* (DW, *n* = 4; DSS, *n* = 6). *P* values are determined by unpaired two-tailed *t* test. (*n*, numbers of mice; mean ± SEM). Red asterisk indicates that most control mice died within 10 days of DSS administration. **d** Images of colon length of male mice at day 7 after 2% DSS administration. WT (DW, *n* = 8; DSS, *n* = 10), *Trim40⁻/⁻* (DW, *n* = 7; DSS, *n* = 7). Graph showing colon

length of mice (WT and *Trim40⁻/⁻*) after DSS administration (right). *P* values are determined by unpaired two-tailed *t* test. (*n*, numbers of mice; mean ± SD). **e** Survival curve of male mice treated with 3.5% DSS for 8 days, followed by normal water for 7 days. WT (*n* = 9), *Trim40⁻/⁻* male mice (*n* = 10). *P* value is determined by log-rank test. (*n*, numbers of mice). **f** Representative H&E staining and IHC staining with E-cadherin of the rectum from WT and *Trim40⁻/⁻* male mice at day 3 after 1% DSS. Scale bars, 100 μm. **g** qPCR showing relative mRNA levels of the indicated genes in 2% DSS-induced colitis. *P* values are determined by Mann–Whitney *U* Test. (*n* ≥ 4 biological independent mice; mean ± SD). *n.s.*; not significant. **h** ROCK1 protein levels in the rectal region from WT and *Trim40⁻/⁻* male mice with 2.5% DSS for 3 days. All data are representative of three independent experiments and source data are provided as a Source Data file.

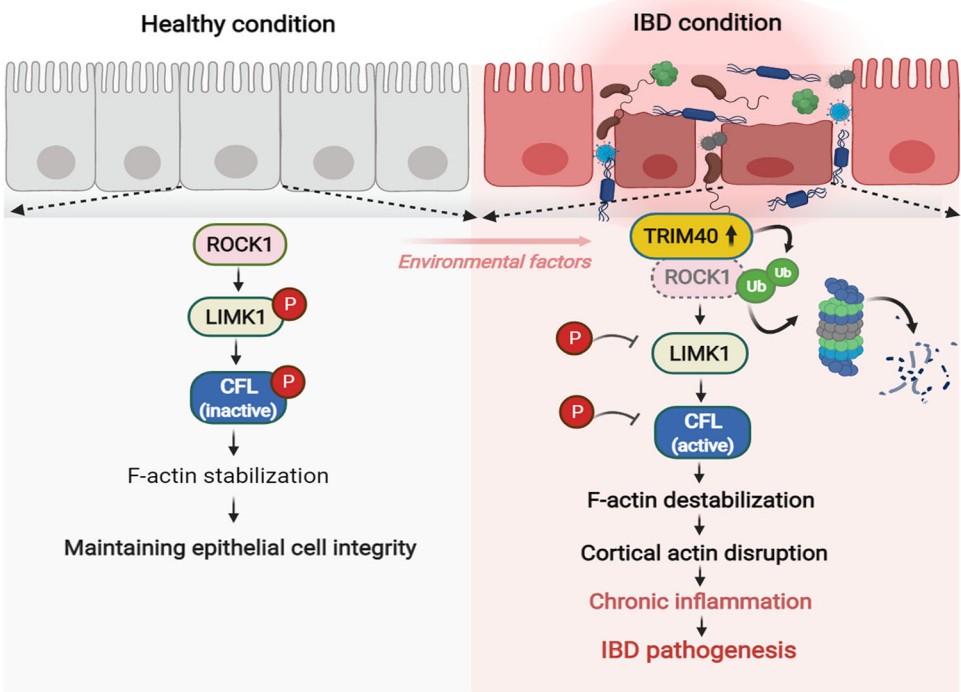

**Fig. 6 | Proposed model of upregulated TRIM40-driven IBD initiation and progression.** The upregulated expression of typically epigenetically silenced *TRIM40* acts as a pathogenic driver of IBD. The proposed mechanism involves the direct targeting of ROCK1 degradation and cortical actin disruption, which accelerates the destruction of epithelial integrity and function, the transition of short- to long-lived inflammation, and ultimately leads to severe intestinal tissue damage (image created with BioRender.com).

epithelial cells, mouse ROCK1 (mROCK1) protein levels were significantly decreased in DSS-treated control mice, attributable to TRIM40 upregulation; however, no differences were observed in mROCK1 expression levels in *Trim40⁻/⁻* mice, regardless of DSS administration (Fig. 5h). Collectively, in vitro and in vivo results suggest that TRIM40 promotes ROCK1 degradation and subsequent F-actin destabilization, resulting in the loss of intestinal epithelial barrier function.

## Discussion

Inflammation is vital for protection and survival during infection or tissue injury. However, the key factors involved in switching an inflammatory response from an appropriate response to an inappropriate chronic response (such as chronic inflammation, manifesting as IBD) remain unclear. Our study proposes that the upregulated expression of epigenetically silenced *TRIM40* acted as a master pathogenic driver of IBD to enable protective-to-pathological inflammation transition, leading to loss of epithelial barrier integrity. The proposed mechanism involves the direct targeting of ROCK1 degradation and the destabilization of the cortical F-actin architecture, which promoted the destruction of epithelial integrity and function, amplified prolonged inflammation, and ultimately led to intestinal tissue damage (Fig. 6). Notably, our results highlighted the resistance to DSS-induced colitis in *Trim40*-deficient mice, supporting the idea that DSS itself is not an essential process of IBD onset. Instead, the activation of otherwise dormant *Trim40* expression and subsequent ROCK1 degradation-driven epithelial barrier disruption were prerequisites for the transition to a long-lasting inflammation. It is plausible why paralogs of *TRIM40* are likely epigenetically silenced in most tissues during evolution, particularly in the intestine. In addition to internal influences, environmental factors have traditionally been considered vital for the onset of IBD[47]. Along similar lines, our results propose that infections enable *TRIM40* to become transcriptionally "open". Therefore, inflammatory shock evoked by various environmental factors, such as infections, high-fat diet, xenobiotic exposure, social-psychological stressors, or vaccinations, may serve as an initial trigger to activate *TRIM40* expression that then leads to chronic pathological inflammation. Thus, our study helps to define a gene-environmental interaction that occurs in IBD onset or progression.

Several studies have proposed an inhibitory role of TRIM40 in antiviral immunity[48–51]; however, these studies have only examined mouse macrophages or fibroblasts derived from mice without including human intestinal cells or tissues or examining epithelial integrity. In particular, RNA-seq analysis showed that *TRIM40* expression was largely silenced in most human tissues, including lymphoid organs from healthy individuals, whereas *TRIM40* was significantly induced in IBD patients, as confirmed by its mRNA and protein levels (Fig. 1b–f). Similar to human tissues, mouse *Trim40* from wild-type mice was also silenced in most tissues, except in the cecum and intestine, where it was expressed at very low levels, compared with intestinal *Trim31* (Supplementary Fig. 7a). Noguchi et al. previously showed that TRIM40 expression is only detectable in rat or mouse IEC-6 or Colon26 cells, respectively, but not in other human intestinal epithelial cell lines (Caco-2, Lovo, HCA-7, WiDr, T84, DLD-1, SW480, HCT116, and COLO201) or other human cell lines[52]. These findings are in agreement with our RNA-seq datasets showing that the mRNA expression of *TRIM40* was barely detectable in most human or mouse tissues and human-derived cell lines, including colonic epithelial cells (HT-29, HCT116, and Caco-2) (Supplementary Fig. 1b and 7a). Given that many paralogous TRIM genes act as oncogenes to promote tumorigenesis, such as *TRIM8, TRIM11, TRIM25, TRIM28, TRIM37,* and *TRIM59,* which are amplified or upregulated in various cancers[23,26,53,54], molecular processes may have evolved over time to silence these potentially harmful paralogous genes.

Interestingly, RhoA activity in TRIM40-overexpressing cells was significantly greater than that in control HT-29 cells (Supplementary Fig. 5g). Previous studies have reported that cell-cell

contact disruption inhibits the function of RhoE, an endogenous RhoA inhibitor, increasing RhoA activity[21,43,55–57]. Thus, TRIM40-mediated disruption of cell-cell junctions may be involved in regulating RhoE function, which potentially promotes RhoA activity through a positive feedback to overcome epithelial cell integrity. In a line with previous reports showing that the YAP is involved in the regulation of Rho-specific guanine nucleotide exchange factor (RhoGEF) and RhoA activity[41,58–60], we also found increased TEAD activation via TRIM40 overexpression without altering β-catenin protein expression levels (Supplementary Fig. 3g, h). Thus, TRIM40 upregulation may directly or indirectly affect YAP/TEAD signaling-dependent RhoGEF or Rho GTPase-activating protein activity, expression, or subcellular localization. Based on our findings, as TRIM40 is involved in the regulation of actin dynamics, its upregulated expression in certain tumor cells or stages may contribute to tumor metastasis.

Several studies have focused on the pathological role of ROCK1 in endothelial cells and intestinal M1 macrophage[61–63]. However, ROCK1 expression has been reported in fibroblasts and epithelial, endothelial, and muscle cells within intestinal tissues[64], indicating that it may play a different role in each cell type. Indeed, our results suggest that TRIM40 overexpression-mediated ROCK1 degradation facilitates F-actin depolymerization and disruption of intestinal epithelial integrity, which occurs during chronic inflammation-induced IBD pathogenesis. ROCK2 inhibition of M1 macrophage polarization has also been proposed, opposite to the effect of ROCK1 signaling, and that it functions as a negative regulator of the immune response by suppressing IRF3 activity[65,66]. This finding provides good insight for further studies on the relationship between TRIM40 overexpression and ROCK2 expression or activity in IBD pathogenesis.

In summary, our study opens up the exciting possibility that targeting specific mechanisms regulating the transition to chronic inflammation driven by the TRIM40-mediated devastation of intestinal homeostasis could represent a strategy for treating IBD. By preventing the activation of *TRIM40* expression or selectively inhibiting its pathological functions, care providers could potentially prevent IBD onset or worsening. Finally, diagnosis remains a challenge in IBD disease management, and our findings suggest that TRIM40 could represent a potential biomarker for use in the diagnosis of both UC and CD using rectum tissue biopsies as well as a therapeutic target to limit IBD initiation and development.

## Methods

### Mice
C57BL/6N wild-type male mice were obtained from the OrientBio and *Trim40⁻/⁻* male mice were generated on C57BL/6N background using CRISPR/Cas9 technique from Cyagen Biosciences. CRISPR guide RNA (gRNA) was used to target exon 1 region in the *Trim40*. Genotypes of mice were investigated using genomic DNA PCR. All mice were maintained in the specific pathogen-free facility of the Yonsei Laboratory Animal Research Center at Yonsei University according to Korean Food and Drug Administration. All mice were housed on a 14-hour light and 10-hour dark cycle with 18–23 °C and 40–60% humidity and were maintained in ventilated cages under standard housing conditions with access to food and water *ad libitum*. No more than 5 mice were housed in cages and all mice used in this study were appropriately monitored daily changes in body weight, food or water intake, and physical appearance. At experimental endpoints, mice were euthanized with $CO_2$ inhalation for at least 5 min until breathing and heartbeat stopped. All animal experiments were reviewed and approved by the Institutional Animal Care and Use Committee of the Yonsei University (YLARC, No. IACUC-202011-1177-01, IACUC-202107-1302-02, IACUC-202107-1302-03, IACUC-202204-1447-01, IACUC-202204-1448-01, IACUC-202209-1541-01 and IACUC-A-202209-1542-01).

### Human patient tissues with inflammatory bowel diseases
Colon biopsy samples for non-IBD, UC, or CD were provided from the Department of Internal Medicine and Institute of Gastroenterology, Yonsei University College of Medicine. Participants, who were patients of Severance Hospital, volunteer in sample collection and informed consent forms were available for all participants and all were 18 years or older at the time of sample collection. This study was approved by the Institutional Review Board of Severance Hospital, Seoul, South Korea (IRB numbers: 4-2012-0680 and 4-2012-0302) and informed consent was obtained from the study participants. IBD cohort included an inflamed biopsy (UC and CD) and non-inflamed biopsy dataset (control). The detailed information of patient samples is provided as a Source Data file.

### Cell lines
Human colonic epithelial cell line HT-29 (HTB-38, ATCC) and HCT 116 (CCL-247, ATCC), and monocytic leukemia cell line U937 (CRL-1593.2, ATCC) cells were cultured in Roswell Park Memorial Institute (RPMI, Welgene) 1640 containing 10% heat inactivated fetal bovine serum (FBS, HyClone) and penicillin-streptomycin (PS, HyClone). Human intestinal epithelial cell line Caco-2 (HTB-37, ATCC), human embryonic kidney (HEK) 293 T (CRL-11268, ATCC), 293 A (R70507, Invitrogen), cervical cancer cell line HeLa (CCL-2, ATCC), lung cancer cell line A549 (CCL-185, ATCC), and breast cancer cell line MDA-MB-231 (CRM-HTB-26, ATCC) were cultured in Dulbecco's Modified Eagle medium (DMEM) supplemented with 10% heat inactivated FBS and PS. The normal colonic epithelial cell line FHC (CRL-1831, ATCC) were cultured in DMEM:F12 supplemented with 10 mM HEPES, 10 ng ml⁻¹ cholera toxin, 0.005 mg ml⁻¹ insulin, 0.005 mg ml⁻¹ transferrin, 100 ng ml⁻¹ hydrocortisone, 10% FBS, and 1% PS. All cells were grown at 37 °C in humidified air with 5% $CO_2$.

### Antibodies and reagents
Primary antibodies used: anti-Myc (Cell Signaling Technology (CST), #2276 and #2278), anti-Ezrin/Radixin/Moesin (ERM) (CST, #3142), anti-p-ERM (CST, #3726), anti-Phospho-Cofilin1 (p-CFL1) (Ser3) (CST, #3313), anti-Profilin1 (PFL1) (CST, #3237), anit-LIMK1 (CST, #3842), anti-p-LIMK1/2 (Thr508/ Thr505) (CST, #3841), anti-Myosin Light Chain 2 (MLC2) (CST, #3672), anti-Phospho-MLC2 (p-MLC2) (CST, #3674), anti-Cofilin1 (CFL1) (abcam, ab11062), anti-Rho-associated coiled-coil-containing protein kinase 1 (ROCK1) (abcam, ab45171; CST, #4035), anti-p21-activated kinase 1 (PAK1) (CST, #2602), anti-β-catenin (Santa Cruz, sc-7963), anti-CD44 (HCAM) (Santa Cruz, sc-7297), anti-vinculin (Santa Cruz, sc-73614), anti-RhoA (Santa Cruz, sc-418), anti-Ubiquitin (Santa Cruz, sc-8017), anti-β-actin (Santa Cruz, sc-47778), anti-α-tubulin (Santa Cruz, sc-23948; Applied Biological Materials (ABM), G094), anti-GAPDH (ThermoFisher, MA5-15738), anti-E-cadherin (CST, #14472; Proteintech, 20874-1-AP), anti-Flag (Novus Biologicals, NBP1-06712; ABM, G191), anti-GFP (Santa Cruz, sc-9996; Roche, #11814460001). The human TRIM40 antibody (Abclon, AC181116-232) was generated in rabbit using synthetic peptides corresponding to epitopes from human TRIM40 (17-26 amino acids). Alexa Fluor 488-, 568-, and 647-conjugated secondary antibodies, Alexa Fluor 568-conjugated Phalloidin (A12380), and Alexa Fluor 594-conjugated Deoxyribonuclease I (D12372) were obtained from Thermo Fisher. Cytochalasin D (C8273), Phorbol 12-myristate 13-acetate (P8139), and lipopolysaccharides (LPS O26:B6, L3755) were obtained from Sigma-Aldrich. Y-27632 2HCl (S1049) and FRAX597 (S7271) were obtained from Selleckchem. Information about antibodies and reagents is listed in Supplementary Data 2.

### Induction of colitis in mice
C57BL/6N wild-type and *Trim40⁻/⁻* male mice at 8- to 10-week old were given 1–2.5% dextran-sulfate-sodium (DSS, w/v 36,000–50,000 Da, MP Biomedicals) in drinking water for 3–7 days. To monitor the survival

rate, mice were administered with 3.5% DSS for 8 days, followed by normal drinking water for 7 days. For the 2,4,6-trinitrobenzene sulfonic acid (TNBS, BIOSYNTH Carbosynth)-induced colitis, 4% TNBS in ethanol was intrarectally administered to 8- to 10-week-old male mice. The control group was also administrated with ethanol using the same technique. The body weight and clinical phenotype of the mice were monitored daily and loss of body weight was represented as the percentage. Mice were sacrificed at indicated time points and collected colon tissues were frozen in liquid nitrogen and used for further analysis.

## LPS injection in mouse

C57BL/6N wild-type male mice (6 weeks of age) were challenged with LPS (O26:B6, L3755). LPS (10 mg kg$^{-1}$ body weight) was diluted in sterile PBS and injected intraperitoneally into mice. Only PBS-injected male mice were used as a control. Mice were monitored for 6 h and then were euthanized for measurement of the expression level of *Trim40*, *Trim31*, and *Il6*.

## Hematoxylin and Eosin (H&E) staining and Immunohistochemistry (IHC)

C57BL/6N wild-type and *Trim40*$^{-/-}$ male mice at 8- to 10-week old were given 1% DSS in drinking water for 3 days, and colon was collected. Collected colon tissues were rolled from the proximal region to rectum by the Swiss roll method and fixed in 10% neutral buffered formalin (NBF). Fixated colon samples were embedded in paraffin and tissue sections were stained with hematoxylin and eosin (H&E). For immunohistochemistry (IHC), antigen retrieved samples were stained with anti-E-cadherin and detected using peroxidase-conjugated secondary antibodies and DAB (3,3′-Diaminobenzidine). Formalin-fixed paraffin-embedded human colon tissues from adjacent normal region of colon cancer patients and IBD patients were stained with anti-TRIM40 antibody (1:1000) and counterstained with hematoxylin. To measure TRIM40 IHC score, we studied five randomly regions for each sample at 100× magnification (range from 0–3).

## In situ hybridization

C57BL/6 N wild-type male mice at 8- to 10-week-old were administrated with 2% DSS in drinking water for 7 days, and colon was then collected and fixed in 10% NBF. Fixed colon tissues were embedded in paraffin for the RNAscope. Single-molecule in situ hybridization was conducted by using Advanced Cell Diagnostic RNAscope 2.5 HD Detection kit (Brown, #322300) according to the manufacturer's instructions. Accession number, target region, and catalog order number for mouse *Trim40* probe is NM_001033235.3, nucleotides 251-1320, and Advanced Cell Diagnostics 1193931-C1, respectively.

## Transfection, retroviral or lentiviral transduction

Cells were transfected with the indicated plasmids using Omicsfect (OmicsBio) in serum free and antibiotic free DMEM for 36–48 h. For preparation of virus particles, HEK293T cells were transfected with plasmids encoding VSV-G and Gag-Pol along with retroviral vector containing a gene of interest. Lentivirus also were produced by transfection in HEK293T cells by using plasmid encoding PAX2, pMD2.G and lentiviral vector containing a gene of interest. After 36-48 h post-transfection, media containing virus particles were harvested and filtered through a 0.45 μm membrane. Cells were transduced with retroviruses and polybrene (4 μg ml$^{-1}$) by centrifugation at 948 × *g* for 45 min and incubated for 4 h at 37 °C in humidified air with 5% CO$_2$. Transduced cells were incubated with fresh media for 24 h, followed by selection with 2 μg ml$^{-1}$ puromycin for 48 h.

## DNA constructs

The human TRIM40 was subcloned into pMSCV (Clontech) vector from human cDNA library. Human ROCK1 and LIMK1 were gifts from Dominic Esposito (Addgene plasmid #70567) and William Hahn & David Root (Addgene plasmid #23511), respectively, and LIMK1 was subcloned into pMSCV vector. The deletion and point mutant constructs of TRIM40 were generated by overlap extension PCR and site-directed mutagenesis, respectively. All constructs were verified by sequencing. All primers and DNA constructs are listed in Supplementary Data 2.

## Immunofluorescence assay

Cells were grown on coverslips and then were fixed with 3.7% formaldehyde in PBS for 5-10 min at room temperature (RT). Fixed cells were permeabilized with 0.2% Triton X-100 in PBS for 5–10 min. After blocking with 2% bovine serum albumin in PBS (PBA) for 0.5–1 h, cells were incubated with the appropriate primary antibody (1:100-1:300) in 2% PBA for 1 h at RT or overnight at 4 °C. Bound antibody was visualized with Alexa Fluor 488-, Alexa Fluor 568, or Alexa Fluor 647-conjugated secondary antibodies in 2% PBA for 0.5–1 h at RT. DAPI (4′,6-diamidino-2-phenylindole, Sigma-Aldrich) was used as a nuclear counterstain. For immunofluorescent staining of F-actin, cells were stained with Alexa Fluor 568-conjugated Phalloidin (1: 300) in 2% PBA for 0.5–1 h at RT. To visualize G-actin, cells were incubated with Alexa Fluor 594-conjugated DNase I (1:100–200) in 2% PBA for 1 h at RT. All confocal images were collected using a LSM 900 confocal laser scanning microscope (Carl Zeiss) and microscopic images were analyzed by the ZEN 3.0 (blue edition) software.

## Microarray and RNA sequencing

Total cellular RNAs were extracted using RNA preparation kits (GeneAll and Enzynomics). To eliminate DNA contamination, we cleaned the extracted RNAs using RNeasy mini kit (Qiagen) and the integrity of RNA was evaluated on 1% agarose gels in Tris-Borate-EDTA (TBE) buffer. Microarray analysis was performed using Affymetrix GeneChip Human Gene 2.0 ST Array. For RNA sequencing, libraries were prepared using the TruSeq Stranded mRNA LT Sample Prep kit according to the manufacture's protocols and sequenced using NovaSeq 6000 System Specifications. Microarray and RNA sequencing were performed three and two times as independent experiments, respectively. The genes that increased or decreased in common were selected and analyzed by using Database for Annotation, Visualization and Integrated Discovery (DAVID) (David.ncifcrf.gov).

## RT-PCR, quantitative PCR (qPCR)

Total cellular RNAs from tissues sample extracted colon tissues of mice and human IBD patients were isolated using RNA preparation kits (GeneAll and Enzynomics) or Trizol followed by chloroform method. Total RNAs (0.5–2 μg) were reverse transcribed with 18 mer oligo (dT) and random hexamer primers using Moloney Murine leukemia Virus (M-MLV) reverse transcriptase (Enzynomics) at 42 °C for 1 h. PCR was performed with appropriate cDNA and primers, and the PCR products were analyzed using ethidium bromide (EtBr)-containing agarose gels. The qPCR reactions were performed using SYBR Green qPCR 2X Premix (Enzynomics) and detected with the Applied Biosystems QuantStudio3 Real-Time PCR System (Thermo Fisher Scientific). The qPCR data were analyzed by the change-in-threshold (CT) values in samples and comparative CT values were normalized by glyceraldehyde-3-phosphate dehydrogenase (GAPDH) mRNA. All used primer sequences are listed in Supplementary Data 2.

## Co-immunoprecipitation and Immunoblot analysis

Cells were lysed with 1% nonyl phenoxypolyethoxylethanol (NP-40, Sigma-Aldrich) in PBS supplemented with protease inhibitor cocktail (Roche) for 0.5–1 h at 4 °C. After centrifugation at 15,928 × *g* for 10 min at 4 °C, cell lysates were incubated with appropriate primary antibodies overnight at 4 °C, and then protein G beads (GenScript) were added to the samples for 1 h at 4 °C. The beads were washed three

times with 1% NP-40. Lysates and immunoprecipitated proteins were denatured with sample buffer or denaturing buffer (50 mM Tris-HCl [pH 6.8], 5% β-mercaptoethanol, and 2% sodium dodecyl sulfate [SDS]) for 7–10 min at 95 °C, respectively. Protein samples were separated by SDS-PAGE and transferred to polyvinylidene difluoride (PVDF) membrane (Millipore). The membranes were blocked with 5% skim milk in PBS containing 0.1% Tween 20 (PBS-T) (BioShop Canada Inc) for 0.5-1 h at RT and incubated with the appropriate primary antibodies for 1 h at RT or overnight at 4 °C. The membranes were washed three times with PBS-T and incubated with horseradish peroxidase (HRP)-conjugated secondary antibodies for 1 h at RT. Bands were visualized using an enhanced chemiluminescence (ECL) detection reagent (Advansta).

### Ubiquitination assay

For TRIM40 mediated ROCK1 ubiquitination assay, cells stably expressing indicated control vector, WT-TRIM40, or TRIM40ΔRING were treated with 10 μM MG132 for 10–12 h. Harvested cells were harvested and lysed in RIPA buffer (50 mM Tris-HCl [pH 8.0], 150 mM NaCl, 1% NP40, 0.1% SDS, 1 mM Ethylenediaminetetraacetic acid [EDTA], 2 mM MgCl$_2$, and 50 mM N-Ethylmaleimide [NEM]) containing protease inhibitors. Cell lysates were immunoprecipitated with anti-ROCK1 antibody overnight at 4 °C and then protein G beads were added to the samples for 1 h at 4 °C. The beads were washed four times with RIPA buffer and eluted by boiling in 1× denaturing buffer for 10 min at 95 °C. Analysis of ROCK1 ubiquitination was performed by immunoblotting with anti-Ubiquitin antibody.

### Rhotekin-RBD pull-down assay

HT-29 cells were washed with PBS, and then lysed with 1% NP-40 on the culture dishes. Active Rho Pull-Down and Detection Kit (Thermo Scientific, 16116) was used to measure the level of active RhoA, according to the manufacturer's instructions. GST-Rhotekin-RBD bound proteins were separated by SDS-PAGE and then immunoblotted with an anti-RhoA antibody.

### Flow cytometry

The expression levels of CD44 and E-cadherin on the cell surface were determined by flow cytometry (FACSCalibur, BD Bioscience). Cells were washed with PBS and then two times with 1% BSA in PBS. Cells were stained with indicated primary antibodies for 1 h at 4 °C. After 1 h, cells were washed two times with 1% BSA in PBS and then incubated with fluorescein isothiocyanate (FITC)-conjugated secondary antibodies for 0.5–1 h for 4 °C in dark, followed by washing in cold PBS. Finally, washed cells were fixed in 1% formaldehyde in PBS and immediately determined by flow cytometry. Samples were analyzed with BD CellQuest Pro software.

### Measurement of trans-epithelial electrical resistance (TEER)

For measurement of epithelial cell integrity, Caco-2 cells ($1 \times 10^5$ cells) were seeded in 24-well transwell plates with polyester membrane filters (pore size 0.4 μm, surface area 1.12 cm$^2$) (Corning). Culture media were added to both apical and basal compartments of inserts and the media were changed every two days for 6-9 days. After cells reach confluence, TEER was monitored by an epithelial volt-ohm meter (Millicell ERS-2, Merck) with STX4 electrodes introduced to the apical and basal surfaces. The TEER values corresponding to the epithelial cell were corrected with background from transwell.

### Soluble or insoluble actin fractionation

HT-29 cells stably expressing control vector or Myc-TRIM40 were harvested and centrifuged $1677 \times g$ for 5 min. Supernatants were discarded and then cell pellets were resuspended in hypotonic lysis buffer (1 mM MgCl$_2$, 2 mM EDTA, 0.5% NP-40, and 2 mM Tris-HCl [pH 6.8]) containing protease inhibitors. After the same volume of hypotonic buffer was added in the resuspended pellets, the cell lysates were

incubated for 10 min at 37 °C in the dark. And then the extracts were vortexed and centrifuged 18,000 × $g$ for 10 min at RT. The supernatants (soluble actin) were transferred to new tube, while the pellets (insoluble actin) were resuspended in hypotonic lysis buffer and then pellets were sonicated. The levels of soluble and insoluble actin were determined by immunoblot analysis.

### Silver staining and mass spectrometry

After co-immunoprecipitation with anti-Myc antibody in HT-29 cells stably expressing control vector or Myc-TRIM40, the eluted proteins were separated by SDS-PAGE and subsequently, gels were visualized by silver staining. The bands of interest were excised from the gels and analyzed by liquid chromatography with tandem mass spectrometry (LC-MS/MS).

### Luciferase assay

293A cells in 12 well plate were transfected with 8 × GTIIC TEAD promoter-luciferase reporter (Addgene plasmid #34615), Renilla luciferase, along with control vector or Myc-TRIM40. After 24 h, cells were lysed and luciferase activity was measured using a Dual-Glo Luciferase Assay System (Promega, E2920). Firefly luciferase activity was normalized to Renilla luciferase activity.

### Surface plasmon resonance (SPR)

The Biacore T200 system (Cytiva) was used to detect the interactions of full length TRIM40 and ROCK1. Full length TRIM40 protein was expressed in *Escherichia coli*, BL21 (DE3) cells and ROCK1 kinase domain (residue 1–415) protein was expressed in *Spodoptera frugiperda* cells (Sf9). The ROCK1 kinase domain protein was immobilized onto the surface of a series S sensor chip CM5 (Cytiva) by amine coupling. For pH scouting, a buffer composed of 10 mM sodium acetate (pH 4.5) was used. The coupling process was carried out based on the molecular weight of ROCK1 kinase domain (MW$_{ligand}$) and full-length TRIM40 (MW$_{analyte}$) to derive the R$_{ligand}$ value (3400RU). The sample was injected (30 μl/min flow rate) in a running buffer composed of HBS-EP$^+$ buffer (10 mM Hydroxyethyl piperazine ethane sulfonicacid [HEPES], 150 mM NaCl, 3 mM EDTA and 0.05% v/v Surfactant P20) at 25 °C. Sensorgrams were recorded and analyzed in real time using Biacore T200 control software. To derive the $K_d$ values, eight different concentrations of full-length TRIM40 were used for binding. All kinetic data were calculated using the Biacore T200 evaluation software.

### Statistical Analysis

All experiments were repeated at least three times with consistent results. Data are presented as mean and SD or SEM. Statistical analyses were performed using GraphPad Prism 8.0 or Microsoft Excel 2019 software. Statistical differences between two means were evaluated with the unpaired two-tailed, Student's $t$ test, log-rank test, Mann–Whitney $U$ Test or fisher's exact test. Differences with $P$-values below 0.05 were considered significant. No samples were excluded from the analysis. The data had a normal distribution and the variance was similar between the groups being statistically compared. No statistical method was used to predetermine sample sizes. The sample size was based on previous experience with experimental variability. The investigators were not blinded to allocation during experiments or outcome assessment. All raw data were provided in Source Data.

### Reporting summary

Further information on research design is available in the Nature Portfolio Reporting Summary linked to this article.

## Data availability

The RNA-seq and microarray data have been deposited in Gene Expression Omnibus (GEO) and are accessible through GEO Series accession numbers GSE199992, GSE200023, and GSE200024. The

mass spectrometry data are available in the ProteomeXchange Consortium via the jPOST partner repository with the accession code identifier PXD038095. The data supporting the findings of this study are available within the article and its Supplementary Information files. Source data are provided with this paper.

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

## Acknowledgements

The authors thank Dr. Shigetsugu Hatakeyama (Hokkaido University Graduate School of Medicine, Hokkaido, Japan) for kindly providing mouse TRIM40 construct. This work was supported by the Samsung Science & Technology Foundation (SSTF-BA1801-08) and the National Research Foundation of Korea (NRF) funded by the Ministry of Science, ICT, and future planning (NRF-2017R1E1A1A01074135, NRF-2022R1A2C3008614, and NRF-2022M3E5F2018597). S.L. was supported by NRF (NRF-2018R1D1A1B07048930) and the National Cancer Center of Korea (NCC-2310480). T.A.L. was supported by NRF (NRF-2019R1A6A3A01096470 and NRF-2020R1I1A1A01072359). J.K., G.P., and H.K. were supported by the Brain Korea (BK21) PLUS Program.

## Author contributions

S.K., J.K., A.P., M.K., W.S., G.P., T.A.L., Y.K., M.K.C., J.H.P., and E.L. performed the biochemical experiments and H.J.K., H.H., and S.L. analyzed RNA-seq and mass spectrometry datasets. H.-S.C., H.W.P., J.H.C., and S.L. provided intellectual input and advised on experimental designs. S.K., S.L., and B.P. designed experiments and wrote the manuscript.

## Competing interests

The authors declare no competing interests.
