## [Peer Review File · Nature Communications]

TRIM40 is a pathogenic driver of inflammatory bowel disease subverting intestinal barrier integrityREVIEWER COMMENTS

Reviewer #1 (Remarks to the Author):

The authors present a well-conducted study on the question of which key factors may drive IBD pathogenesis. By analyzing gene expression datasets, they found that the E3 ligase TRIM40 is statistically upregulated in IBD patients. While TRIM40 may be involved in IBD by targeting multiple substrates, the authors validated RIOK1 as a key ubiquitination substrate of TRIM40 in the context of epithelial integrity. The experimental results are quite convincing and overall support the conclusions.

In Figure 2, the authors showed that overexpression of TRIM40 impairs a series of molecular processes, which are important for the function of F-actin. Given that TRIM40 is an E3 ligase, it would be interesting to test whether an E3 ligase-dead mutant still affects these molecular processes. Deletion of the whole RING domain in at least some TRIM family proteins can affect their cellular distributions. A minimal mutation, which would only change one or two conserved Cys residues in the RING domain and therefore become a complete ligase-dead form of TRIM40, may be more suitable for the above experiment.

Reviewer #2 (Remarks to the Author):

The actin cytoskeleton plays a critical role in maintaining intestinal epithelial barrier function and, disruption of this architecture results in a leaky barrier that contributes to pathogenesis of chronic inflammatory diseases.

This study reports that a subset of inflammatory bowel disease (IBD) patients have increased expression of tripartite motif-containing protein 40 (TRIM40), an E3 ligase that directly targets ROCK1 resulting in destabilization of cortical actin and loss of epithelial barrier function. This is an interesting study with clinically relevant findings related to actin destabilization during IBD. Some concerns are listed below:

Major concerns:

- Most of the in vitro work was performed in cell lines overexpressing TRIM40. In addition, murine CT-26 cells have a fibroblast phenotype. Therefore, it would be beneficial to replicate key findings in primary intestinal epithelial cells isolated from WT and TRIM40 KO mice.
- It would be useful to demonstrate effects of DSS or pro-inflammatory cytokines on expression levels of TRIM40 in colonic epithelial cells in vivo by RNA in situ hybridization assay (RNAscope).
- In Fig.1b lack of expression of TRIM40 in normal tissues from a range of organs is compared to expression in ulcerative colitis and Crohn's disease. A better comparison would be to show expression

levels of TRIM40 during inflammation or disease in other organs compared to normal tissue.

- Does forced overexpression of TRIM40 in IECs result in altered cell migration responses?
- The mechanism by which TRIM40 expression alters F-actin and thereby junctional integrity in epithelial cells needs to be better defined. The authors show that TRIM40 selectively downregulates ROCK1 thereby leading to F-actin depolymerization. However, it is not clear that this effect on F-actin is independent of PAK. It is known that LIMK1 is activated by both ROCK1 and PAK. The authors should therefore show the effect of TRIM40 OE on PAK1 protein levels.
- Previous work has shown that ROCK1 mediates phosphorylation of MLC2. Therefore the authors should determine if levels of MLC2 phosphorylation are altered in cells overexpressing TRIM40? In addition, the authors should co-stain cells for pMLC2 and F-actin to better visualize changes in the organization of actomyosin structures at the cortex and cell-cell junctions.
- It would be useful to determine changes in RhoA activity in cells overexpressing TRIM40. This can be done using an active Rho biosensor or an active Rho probe to detect the activity of RhoA by either live cell imaging or rGBD pulldown assay.

Minor concerns:

- There is a previous study that shows that TRIM40 is highly expressed in normal gastrointestinal tissues with reduced expression observed in chronic inflammatory lesions (PMID 21474709). This work should be discussed in the context of the author's findings.
- Given the lack of expression in inflamed ileal tissues, is TRIM40 signaling only relevant to rectal inflammation? Given that a UC group is included in the ileal graph, there should be a Crohn's disease group in the rectal mucosa graph in Fig. 2C.
- In Figure 2 Vinculin staining in control cells appears to be only at the focal adhesions. The authors should compare junctional localization of vinculin in control cells to cells expressing TRIM40?
- The authors should include a new blot for pERM proteins and consider doing densitometry analyses (Fig. 2D).
- Throughout the manuscript, the authors refer to overexpression of TRIM40 as a "gain-of-function". Is there data showing OE of TRIM40 leads to increased activity of TRIM40?
- What is the level of knockdown/knockout of TRIM40 in TRIM40 KO mice?
- TRIM Extended data 2 Figure 2: Authors should improve the contrast and resolution of the images selected.

Reviewer #3 (Remarks to the Author):

The data presented by Kang et al. demonstrated that TRIM40 expression in inflamed intestinal epithelia cells contribute to barrier dysfunction, and is therefore relevant on the context of chronic intestinal inflammation, such as in IBD. Actually, gene and protein expression of TRIM40 is significantly upregulated in human and mouse colitis. A detailed in vitro study using several cell lines enabled the authors to describe the mechanism by which expression of TRIM40 leads to degradation of ROCK and alterations of cell cytoskeleton. Furthermore, the physiological relevance of these findings is highlighted by the impressive protection observed in TRIM40 KO mice upon treatment with DSS. I believe these data are novel and might contribute to the identification of biomarkers and/or therapy strategies in the context of epithelial restitution in IBD. However, some discrepancies with available literature, as well as some important aspects should be addressed before the manuscript can be accepted for publication. Some of these aspects are described here:

- Most of the TRIM40 expression data in intestinal tissue shown in the manuscript are obtained from publically available datasets (gene expression); while protein data are based on the detection of c-Myc. Although all the data are convincing, I think addressing two additional aspects might contribute to the scientific quality of the manuscript. First, it would be optimal if some of the main findings regarding TRIM40 protein expression are verified using a specific anti-TRIM40 antibody. Moreover, and using this technical approach, I believe the manuscript will benefit from an initial analysis of TRIM40 expression within the gut tissue, both mouse and human. Probably it would be also good to have comparison between TRIM40 and other TRIM proteins, such as TRIM69 or TRIM31, as mentioned throughout the manuscript. In fact, even in Fig. 1b, it is shown that there is some low expression of TRIM40, so is this relevant? where this expression comes from? epithelium? Does this have a relevance in the context of intestinal niches, colon versus small intestine?
- The publication from Noguchi, 2011 (PMID: 21474709) described contradictory observations regarding the expression pattern profile of TRIM40 in gastrointestinal tissue, and upon inflammation and/or cancer. This publication has to be mentioned and discussed, at least; or even proven that the mechanism suggested can be discarded in those samples included in this study. Based on that publication, is there a different behaviour affecting TRIM40 expression in case of inflammation vs. cancer? If this is the case, I think a non tumor cell line should be analysed, or even intestinal organoids to confirm the mechanism proposed by the authors.
- Concerning the filamentous patterns in the cytoplasm (ED Fig. 2a): are those membrane protrusions???
- Nuclear localization of TRIM40 should be discussed
- As mentioned in my first comment, expression of TRIM40 using a specific antibody would be informative. This also applied to the expression in Control-HT29, and the validation of the main findings with a specific anti-TRIM40 antibody.
- CytoD induced a different pattern than TRIM40 overexpression and/or PMA, as indicated in ED Fig. 3d.

I'm not so sure about cortical actin retraction (Fig.2a), and this is simply a matter of different cell shape. This should be discussed in the text.

- I think the data concerning TEER lack relevance, since there is a clear issue concerning cell adherence. do not form a monolayer; it would then be important to show that TRIM40 overexpressing cells used for TEER measurement reach confluence. Otherwise, the TEER data should not be shown.
- Relevance of LPS injected mouse model; if the authors considered that this is a relevant observation, the n number should be increased. What is the added value of these data?
- I definitely think that this study would benefit from data generated with TRIM40KO mice subjected to other colitis model where epithelial integrity is important.
- Some publications described ROCK expression upregulation in IBD; how this can be explained based on the present data; please, discuss.
- Abstract: revise the sentence about the different mutants; was it only Δ RING-mut the one leading to recovery of ROCK1 and actin cytoskeleton; as it reads it seems that all three mutants affected ROCK1 degradation and actin cytoskeleton.

Minor

- L81: its expression is in IBD samples...
- Figure 1 should be organized in a way that the subfigures follow an order; otherwise, it is complicated to read the figure
- L88-89: the sentence "this notable upregulation in the expression of the otherwise epigenetically silenced TRIM40 occurred at a high frequency across a significant distribution of patients with IBD."; is this based on the data shown in Fig 1.d. I think this is overstated, and I would consider rephrasing it to show that in this a rather small cohort you can see an upregulation of the expression.
- ED Fig. 2b: pictures from Caco2 are not visible; I would suggest choosing other representative pictures.
- L252-253: what does it mean DSS alone or DSS in conjunction with inflammation???
- L268: epithelial barrier
- ED 7d: picture from distal colon, WT DSS, is it from the same magnification as the other pictures???
- L300-301: DSS accompanying inflammation itself is not an essential process in IBD: what does this sentence mean?

Comments from the reviewers that required a response are in bold italics, with each reply appearing in normal font just below the comment. In the replies, new material is emphasized in bold.

Reviewer #1 (Remarks to the Author):

The authors present a well-conducted study on the question of which key factors may drive IBD pathogenesis. By analyzing gene expression datasets, they found that the E3 ligase TRIM40 is statistically upregulated in IBD patients. While TRIM40 may be involved in IBD by targeting multiple substrates, the authors validated RIOK1 as a key ubiquitination substrate of TRIM40 in the context of epithelial integrity. The experimental results are quite convincing and overall support the conclusions.

In Figure 2, the authors showed that overexpression of TRIM40 impairs a series of molecular processes, which are important for the function of F-actin. Given that TRIM40 is an E3 ligase, it would be interesting to test whether an E3 ligase-dead mutant still affects these molecular processes. Deletion of the whole RING domain in at least some TRIM family proteins can affect their cellular distributions. A minimal mutation, which would only change one or two conserved Cys residues in the RING domain and therefore become a complete ligase-dead form of TRIM40, may be more suitable for the above experiment.

The reviewer raises an important point about the possible artefactual effects of deletion mutants lacking the entire RING domain. Following previous reports that an E3 ligase-dead TRIM40 mutant with serine in the place of cysteine 29 in the RING domain (TRIM40-C29S) lost its functional activity (Chunyuan Zhao et al., *Cell Reports*, 2017), we generated an N-terminal Myc-tagged TRIM40-C29S mutant and examined its effect on ROCK1 degradation and cortical actin disruption. Consistent with our previous results with a TRIM40 mutant lacking the whole RING domain, immunoblot and confocal microscopy analyses showed that TRIM40-C29S not only failed to degrade ROCK1 but also attenuated cortical actin disruption. We have incorporated these updated results into the **new Fig. 4d** and **4e** of the revised manuscript.

Comments from the reviewers that required a response are in bold italics, with each reply appearing in normal font just below the comment. In the replies, new material is emphasized in bold. We would also like to note that the order of some questions was changed without any omission to properly address the reviewer's questions.

Reviewer #2 (Remarks to the Author):

The actin cytoskeleton plays a critical role in maintaining intestinal epithelial barrier function and, disruption of this architecture results in a leaky barrier that contributes to pathogenesis of chronic inflammatory diseases.

This study reports that a subset of inflammatory bowel disease (IBD) patients have increased expression of tripartite motif-containing protein 40 (TRIM40), an E3 ligase that directly targets ROCK1 resulting in destabilization of cortical actin and loss of epithelial barrier function. This is an interesting study with clinically relevant findings related to actin destabilization during IBD. Some concerns are listed below:

Major concerns:

• Most of the in vitro work was performed in cell lines overexpressing TRIM40. In addition, murine CT-26 cells have a fibroblast phenotype. Therefore, it would therefore be beneficial to replicate key findings in primary intestinal epithelial cells isolated from WT and TRIM40 KO mice.

The reviewer raised a valid point by requesting the inclusion of additional experiments to show the role of TRIM40 in mouse primary intestinal epithelial cells. To accomplish this, we tried to obtain primary intestinal epithelial cells from DSS-treated wild-type mice throughout the more than four months that we spent working on this revision. Disappointingly, we were unable to collect sufficient numbers of intestinal epithelial cells from DSS-treated wild-type mouse colon tissues for immunoblot analysis, probably because of severe tissue damage caused by the DSS treatment. This might also be the reason that was hard to find experimental procedures referenced in published papers for purifying primary intestinal cells from DSS-induced mice. We were, however, able to collect adequate numbers of epithelial cells from *Trim40*-deficient mice even with DSS treatment, because those mice are resistant to DSS-induced colitis (Fig. 5c-g).

Therefore, to provide more convincing results, we replicated key findings in FHC cells, a normal colonic epithelial cell line, as the reviewer has suggested. Consistent with our previous results, TRIM40 overexpression in FHC cells caused degradation of endogenous ROCK1 and subsequent dephosphorylation of CFL1, resulting in disruption of cortical actin formation. We have incorporated these new results into the **new Fig. 3c** and removed the previous results with CT-26 cells, as the reviewer pointed out that murine fibroblast CT-26 cells are not adequate to clarify the pathological effect of TRIM40 in IBD.

• It would be useful to demonstrate effects of DSS or pro-inflammatory cytokines on expression levels of TRIM40 in colonic epithelial cells in vivo by RNA in situ hybridization assay (RNAscope).

Per the reviewer's suggestion, we examined the upregulation of *Trim40* expression in colon tissues from DSS-induced mice by performing an RNA *in situ* hybridization assay. We clearly observed that *Trim40* expression was significantly increased in DSS-treated mouse colon (**new Fig. 5b** and **Supplementary Fig. 7c**). Based on these new results and our previous results from public RNA-seq and qPCR data with DSS-induced mice (Fig. 1, Fig. 5, and Supplementary Fig. 7), we are now confident that *Trim40* expression was upregulated in the rectum and distal colon of mice that were administered DSS. We have provided the new data in the **new Fig. 5b** and **Supplementary Fig. 7c** of the revised manuscript.

• ***In Fig.1b lack of expression of TRIM40 in normal tissues from a range of organs is compared to expression in ulcerative colitis and Crohn's disease. A better comparison would be to show expression levels of TRIM40 during inflammation or disease in other organs compared to normal tissue.***

The reviewer's comment highlights the important issue of clarifying the *Trim40* upregulation in UC and CD in comparison with normal colon tissue. We previously showed that *Trim40* expression was much higher in IBD samples than in intestinal tissues from healthy controls (Fig. 1b, bold letters of the revised manuscript). To avoid confusion, we modified the representations of letters of 'Colon' and 'Small intestine' in the graph with normal tissues, as highlighted by a bold, and we included the additional information in the Figure legends of the revised manuscript to make them clearer.

To provide more convincing results, we also performed immunohistochemistry (IHC) assays with anti-human TRIM40 antibody using colon tissues from controls or IBD patients. We found that TRIM40 proteins were mostly distributed throughout the epithelial cells of the entire colon (**new Fig. 1f**). Furthermore, TRIM40 protein expression was barely detectable in healthy controls, whereas it was much higher in colon tissues of patients with UC or CD (**new Fig. 1f**), which is consistent with the RNA-seq and qPCR results previously shown in Fig. 1a-e. We also examined the upregulation of *Trim40* expression in colon tissues from DSS-induced colitis in mice by performing an RNA *in situ* hybridization assay (ISH). We clearly observed that *Trim40* expression was significantly increased in DSS-treated mouse colon (**new Fig. 5b** and **Supplementary Fig. 7c**).

Based on these new results and our previous results from public RNA-seq data and qPCR with DSS-induced mice (Fig. 1, Fig. 5, and Supplementary Fig. 7), we are now confident that *Trim40* expression was upregulated in the rectum and distal colon of mice that were administered DSS. We incorporated the new human IHC datasets into the **new Fig. 1f** and provided the new ISH datasets in the **new Fig. 5b** and **Supplementary Fig. 7c** of the revised manuscript.

• ***The mechanism by which TRIM40 expression alters F-actin and thereby junctional integrity in epithelial cells needs to be better defined. The authors show that TRIM40 selectively downregulates ROCK1 thereby leading to F-actin depolymerization. However, it is not clear that this effect on F-actin is independent of PAK. It is known that LIMK1 is activated by both ROCK1 and PAK. The authors should therefore show the effect of TRIM40 OE on PAK1 protein levels.***

The reviewer raised an important point about the effect of TRIM40 overexpression on PAK1 expression levels. We performed immunoblot analysis with anti-PAK1 antibody to assess endogenous PAK1 levels; however, TRIM40-overexpressing HT-29 cells did not show any considerable difference in PAK1 expression. Therefore, we conclude that TRIM40 selectively degrades ROCK1, but not PAK1, leading to blockade of LIMK1 phosphorylation. These new data were incorporated into **Supplementary Fig. 5c** of the revised manuscript.

• Previous work has shown that ROCK1 mediates phosphorylation of MLC2. Therefore, the authors should determine if levels of MLC2 phosphorylation are altered in cells overexpressing TRIM40? In addition, the authors should co-stain cells for pMLC2 and F-actin to better visualize changes in the organization of actomyosin structures at the cortex and cell-cell junctions.

The reviewer's comment highlights the important issue of clarifying the effect of TRIM40 overexpression on MLC2 phosphorylation. We showed that TRIM40 overexpression blocks the phosphorylation of both LIMK1 and MLC2 in Fig. 3b of the original manuscript. (Please also see **Fig. #1** at the right panels, red box)

To confirm these results, we additionally examined the subcellular patterns of MLC2 phosphorylation and F-actin in TRIM40-overexpressing cells. Phosphorylated MLC2 failed to colocalize with cortical F-actin in TRIM40-overexpressing HT-29 cells, which is compatible with the immunoblot results showing the failure of MLC2 phosphorylation (Fig. 3b and **new Supplementary Fig. 5f**). These results demonstrate that TRIM40 acts to selectively downregulate ROCK1, preventing a series of phosphorylation events, including LIMK1, MLC2, and CFL1 phosphorylation, that are critical for facilitating F-actin formation and stabilization. We incorporated these new results in the **new Supplementary Fig. 5f** of the revised manuscript.

• It would be useful to determine changes in RhoA activity in cells overexpressing TRIM40. This can be done using an active Rho biosensor or an active Rho probe to detect the activity of RhoA by either live cell imaging or rGBD pulldown assay.

The reviewer raised a valid point by asking if TRIM40 overexpression affects RhoA activity even if there is no change in the RhoA expression level. As the reviewer suggested, we performed an rGBD pulldown assay to assess RhoA activity in control and TRIM40-overexpressing HT-29 cells. Intriguingly, the activity of RhoA in TRIM40-overexpressing cells was significantly greater than that in control HT-29 cells (**new Supplementary Fig. 5g**). Previous studies reported that disruption of cell-cell contact inhibits the function of RhoE, an endogenous RhoA inhibitor, leading to increased

RhoA activity (Pierre Chardin, *Mol. Cell Biol.*, 2006; Catherine F. Cowell et al., *Journal of Cellular Biochemistry*, 2009; Cristina Hidalgo-Carcedo et al., *Nat. Cell Biol.*, 2011; Takuya Kato, *Cell Reports*, 2014, Richard G. Hodge and Anne J. Ridley, *Mol. Cell Biol.*, 2016). Therefore, TRIM40-mediated disruption of cell-cell contact may be involved in regulating RhoE function, which potentially promotes RhoA activity by acting as positive feedback to overcome epithelial cell integrity. In a line with the previous reports showing the involvement of YAP/TAZ signaling in RhoGEF or RhoA activation (Totaro et al., *Nat. Cell Biol.*, 2018; Panciera et al., *Mol. Cell Biol.*, 2017; Dupont et al., *Nature*, 2011; Qiao et al., *Cell Reports*, 2017), we also found an increase in TEAD activation by TRIM40 overexpression without altering β -catenin protein expression levels (Supplementary Fig. 3g and h), possibly that TRIM40 may directly or indirectly affect YAP/TEAD signaling-dependent RhoGEF or Rho GTPase-activating protein (RhoGAP) activity, expression, or subcellular localization. We therefore included additional statements in the Discussion section of the revised manuscript to reflect the reviewer's points.

• **Does forced overexpression of TRIM40 in IECs result in altered cell migration responses?**

(Although this question was originally positioned in the front of the point-by-point response, it was moved here to avoid a redundant explanation.)

The reviewer raised a valid point by asking if TRIM40 overexpression affects cell migration. We examined the effect on cell migration in Myc-tagged TRIM40-overexpressing HT-29 cells by conducting a transwell migration assay. Intriguingly, TRIM40 overexpression resulted in a significant increase in epithelial cell migration. As mentioned above, TRIM40 overexpression-mediated disruption of cell-cell junctions may alter RhoE function and thus maximize RhoA activity to overcome epithelial cell integrity, which might promote FAK/paxillin activity for cell migration. (Please see the additional **Fig. #1** at the bottom.)

This is a good insight for future studies of the relationship between TRIM40 and cell migration or metastasis, so we would like to include this result only in the point-by-point response for the current manuscript. We included additional statements in the Discussion section of the revised manuscript to reflect the reviewer's points.

* **Note to Figures:** (Left) Representative images and statistical results of Transwell migration assays of Mock or TRIM40-overexpressing HT-29 cells, *** $P < 0.001$ (Student's t -test). Scale bars, 100 μ m. (Right) Immunoblots showing TRIM40 expression levels in cells used in Transwell migration assay.

Minor concerns:

• **There is a previous study that shows that TRIM40 is highly expressed in normal gastrointestinal tissues with reduced expression observed in chronic inflammatory lesions (PMID 21474709). This**

work should be discussed in the context of the author's findings.

As the reviewer mentioned, *Noguchi et al.* previously showed TRIM40 expression levels in various mouse tissues and human/mouse cell lines. Although it was difficult to accurately compare relative TRIM40 expression patterns between different mouse tissues because of unequal amounts of protein samples across wells, **mouse TRIM40** is expressed strongly in the small intestine and moderately in stomach, cerebellum, heart, and testis. (Please see the **red boxes** of the additional **Fig. #1A** at the end of this response.) They have also checked the TRIM40 expression levels in various human and mouse cell lines and shown that TRIM40 expression is **detectable in rat/mouse IEC-6 or Colon26** cells but not in nine different **human** intestinal epithelial cell lines (Caco2, Lovo, HCA-7, WiDr, T84, DLD-1, SW480, HCT116, and COLO201) nor in other human cell lines. (Please see the blue or red boxes of the additional **Fig. #1B** at the end of this response.) These findings are in agreement with our RNA-seq datasets showing that the mRNA expression of *TRIM40* was barely detectable in most human tissues and human-derived cell lines, including colonic epithelial cells (HT-29, HCT116, and Caco-2). (Fig. 1b and **Supplementary Fig. 1b**; please also see the additional **Fig. #2** at the end of this response.) In addition, we have shown that mouse TRIM40 is expressed in mouse intestine and colon samples, albeit at a very low level compared with TRIM31. (**Supplementary Fig. 7a**; please also see the additional **Fig. #2** at the end of this response.) We included additional statements in the Discussion section of the revised manuscript to reflect the reviewer's points.

Fig. #1 (Figs. 1A and 1B from the published data) [Keita Noguchi et al., *Carcinogenesis*, 2011]

Fig. #2 (in Fig. 1b of the revised manuscript)

(in Supplementary Fig. 1b of the revised manuscript)

(in Supplementary Fig. 7a of the revised manuscript)

• Given the lack of expression in inflamed ileal tissues, is TRIM40 signaling only relevant to rectal inflammation? Given that a UC group is included in the ileal graph, there should be a Crohn's disease group in the rectal mucosa graph in Fig. 2C.

We agree with the reviewer's point including TRIM40 expression patterns in CD in the rectal mucosa graph. Although the public RNA datasets with rectum mucosa from UC patients were massive and easily accessible, we could not find any RNA datasets with CD patients. This is probably because CD can sporadically manifest throughout the whole gastrointestinal tract and is not limited to the colon or rectum, whereas UC specifically affects parts of the distal colon and rectum. To avoid confusion, we modified the order of graphs and included detailed information in the text of the revised manuscript to reflect the reviewer's point.

• In Figure 2 Vinculin staining in control cells appears to be only at the focal adhesions. The authors should compare junctional localization of vinculin in control cells to cells expressing TRIM40?

We agree with the reviewer's comment. To provide a clearer image, we repeated this experiment several times; however, vinculin was barely detectable in the region of cell-cell contact and was mostly distributed at focal adhesions in HT-29 cells, which is consistent with the previous results. According to previous reports (Shoko Ito et al., *Nature Communications*, 2017; Sousa-squiavinato et al., *Molecular Cell Research*, 2019), vinculin localizes mainly at focal adhesions in HT-29 cells, but it translocates and accumulates at cell-cell junctions when junctional reorganization occurs in cells treated with microtubule polymerization inhibitors such as nocodazole. Similarly, we observed that the majority (> 80~90%) of vinculin localized near focal adhesions, with small numbers of cells showing vinculin staining at cell-cell contact regions under normal conditions. To avoid confusion, we provided more convincing images representing vinculin in Fig. 2b and included additional statements in the text of the revised manuscript to reflect the reviewer's point.

• The authors should include a new blot for pERM proteins and consider doing densitometry analyses (Fig. 2D).

We agree with the reviewer's comments and tried to perform the experiment many times to achieve more convincing results. Disappointingly, it was hard to obtain clear immunoblots with anti-pERM antibody, probably due to the poor antibody quality or because blockade of ERM phosphorylation was rapidly or transiently induced by TRIM40. To resolve this challenge, we established a Tet-On system for doxycycline-inducible TRIM40 expression, allowing us to highly overexpress TRIM40 in a controlled manner and accurately monitor the specific effect of TRIM40 overexpression in HT-29 cells. Consistent with our previous results, we observed that the level of phosphorylated ERM was significantly decreased in HT-29 cells after 48h of doxycycline treatment, while the expression of TRIM40 was maximized. We replaced the previous immunoblot datasets with the new datasets produced with the Tet-On system and incorporated the new results into the **new Fig. 2d-f** and also revised the text of the manuscript to reflect the new results.

- ***Throughout the manuscript, the authors refer to overexpression of TRIM40 as a “gain-of-function”. Is there data showing OE of TRIM40 leads to increased activity of TRIM40?***

We agree with the reviewer's comments. To avoid confusion, we reworded this term in the revised manuscript.

- ***What is the level of knockdown/knockout of TRIM40 in TRIM40 KO mice?***

We thank the reviewer for catching this lack of information. We have incorporated additional results into the **new Fig. 5g** (last graph) showing the TRIM40 mRNA expression levels in the colon tissue from wild-type and *Trim40*-deficient mice.

- ***TRIM Extended data 2 Figure 2: Authors should improve the contrast and resolution of the images selected.***

We agree with the reviewer's comment. To clarify the cellular morphologic difference between control and TRIM40-expressing cells, we provided clearer and sharper images showing that TRIM40 overexpression facilitates aberrant morphological changes with cell-to-cell repulsion and significant gaps. We incorporated these results into the **new Supplementary Fig. 2b** of the revised manuscript.

Comments from the reviewers that required a response are in bold italics, with each reply appearing in normal font just below the comment. In the replies, new material is emphasized in bold.

Reviewer #3 (Remarks to the Author):

The data presented by Kang et al. demonstrated that TRIM40 expression in inflamed intestinal epithelia cells contribute to barrier dysfunction, and is therefore relevant on the context of chronic intestinal inflammation, such as in IBD. Actually, gene and protein expression of TRIM40 is significantly upregulated in human and mouse colitis. A detailed in vitro study using several cell lines enabled the authors to describe the mechanism by which expression of TRIM40 leads to degradation of ROCK and alterations of cell cytoskeleton. Furthermore, the physiological relevance of these findings is highlighted by the impressive protection observed in TRIM40 KO mice upon treatment with DSS. I believe these data are novel and might contribute to the identification of biomarkers and/or therapy strategies in the context of epithelial restitution in IBD. However, some discrepancies with available literature, as well as some important aspects should be addressed before the manuscript can be accepted for publication.

Some of these aspects are described here:

• Most of the TRIM40 expression data in intestinal tissue shown in the manuscript are obtained from publically available datasets (gene expression); while protein data are based on the detection of c-Myc. Although all the data are convincing, I think addressing two additional aspects might contribute to the scientific quality of the manuscript. First, it would be optimal if some of the main findings regarding TRIM40 protein expression are verified using a specific anti-TRIM40 antibody. Moreover, and using this technical approach, I believe the manuscript will benefit from an initial analysis of TRIM40 expression within the gut tissue, both mouse and human. Probably it would be also good to have comparison between TRIM40 and other TRIM proteins, such as TRIM69 or TRIM31, as mentioned throughout the manuscript. In fact, even in Fig. 1b, it is shown that there is some low expression of TRIM40, so is this relevant? where this expression comes from? epithelium? Does this have a relevance in the context of intestinal niches, colon versus small intestine?

The reviewer raised an important point by requesting the measurement of TRIM40 protein expression using a specific anti-TRIM40 antibody. As the reviewer suggested, we performed immunohistochemistry (IHC) assays with anti-human TRIM40 antibody using colon tissues from non-IBD control individuals and patients with IBD. We found that TRIM40 proteins were mostly distributed throughout the epithelial cells of the entire colon. Furthermore, TRIM40 protein expression was barely detectable in non-IBD controls, whereas it was much higher in colon tissues of patients with UC or CD, which is consistent with the RNA-seq results previously shown in Fig. 1. Although we would like to conduct additional IHC analyses in mouse colon tissues per the reviewer's comment, there was not a commercial antibody available for mouse TRIM40 to use in IHC, IFA, or immunoblot assays. Instead, to confirm these results, we examined the upregulation of *Trim40*

expression in colon tissues from DSS-induced mice by performing an RNA *in situ* hybridization assay (ISH). We clearly observed that *Trim40* expression was significantly increased in DSS-treated mouse colon (**new Fig. 5b** and **Supplementary Fig. 7c**).

Based on these new results and our previous results from public RNA-seq data and qPCR with DSS-induced mice (**Fig. 1** and **Fig. 5**), we are now confident that *Trim40* expression was upregulated in the rectum and distal colon of mice that were administered DSS. We incorporated the new human IHC datasets into the **new Fig. 1f** and provided the new ISH datasets in the **new Fig. 5b** and **Supplementary Fig. 7c** of the revised manuscript.

• The publication from Noguchi, 2011 (PMID: 21474709) described contradictory observations regarding the expression pattern profile of TRIM40 in gastrointestinal tissue, and upon inflammation and/or cancer. This publication has to be mentioned and discussed, at least; or even proven that the mechanism suggested can be discarded in those samples included in this study. Based on that publication, is there a different behaviour affecting TRIM40 expression in case of inflammation vs. cancer? If this is the case, I think a non tumor cell line should be analysed, or even intestinal organoids to confirm the mechanism proposed by the authors.

As the reviewer mentioned, *Noguchi et al.* previously showed TRIM40 expression levels in various mouse tissues and human/mouse cell lines. Although it was difficult to accurately compare relative TRIM40 expression patterns between different mouse tissues because of unequal amounts of protein samples across wells, mouse TRIM40 is expressed strongly in the small intestine and moderately in stomach, cerebellum, heart, and testis. (Please see the red boxes of the additional **Fig. #1A** at the end of this response.) They have also checked the TRIM40 expression levels in various human and mouse cell lines and shown that TRIM40 expression is detectable in rat/mouse IEC-6 or Colon26 cells but not in nine different human intestinal epithelial cell lines (Caco2, Lovo, HCA-7, WiDr, T84, DLD-1, SW480, HCT116, and COLO201) nor in other human cell lines. (Please see the red or blue boxes of the additional **Fig. #1B** at the end of this response.) These findings are in agreement with our RNA-seq datasets showing that the mRNA expression of *TRIM40* was barely detectable in most human tissues and human-derived cell lines, including colonic epithelial cells (HT-29, HCT116, and Caco-2). (**Fig. 1b** and **Supplementary Fig. 1b**; please also see the additional **Fig. #2** at the end of this response.) In addition, we have shown that mouse TRIM40 is expressed in mouse intestine and colon samples, albeit at a very low level compared with TRIM31. (**Supplementary Fig. 7a**; please also see the additional **Fig. #2** at the end of this response.) Nevertheless, we agree with the reviewer's point about the possibility of different effects of TRIM40 in IBD pathogenesis and tumorigenesis. Therefore, we performed a reciprocal immunoblot analysis of TRIM40-mediated ROCK1 degradation and phospho-CFL1 levels in FHC cells, a non-tumor colon epithelial cell line. Consistent with the results in HT-29 cells, we found that TRIM40 overexpression clearly degrades ROCK1 and subsequently disrupts CFL1 phosphorylation in FHC cells. We incorporated these new results into the **new Fig. 3c** and included additional statements in the Discussion section of the revised manuscript to reflect the reviewer's points.

Fig. #1 (Figs. 1A and 1B from the published data) [Keita Noguchi et al., *Carcinogenesis*, 2011]

Fig. #2 (in Fig. 1b of the revised manuscript)

(in Supplementary Fig. 1b of the revised manuscript)

(in Supplementary Fig. 7a of the revised manuscript)

• Concerning the filamentous patterns in the cytoplasm (ED Fig. 2a): are those membrane protrusions???

The reviewer raised an interesting question by asking if cytosolic filamentous patterns are membrane protrusions in images of TRIM40-overexpressing HT-29 cells. Previous studies showed that cells depleted of ROCK1 or treated with the ROCK1 inhibitor Y-27632 exhibited increased integrin-mediated cell adhesion, cell elongation, membrane protrusion, and cancer cell migration and invasion (Rebecca A. Worthylake and Keith Burridge, *The Journal of Biological Chemistry*, 2003; Michele A. Wozniak et al., *Molecular Biology of the Cell*, 2005; Francisco M. Vega et al., *The Journal of Cell Biology*, 2011; V Harma et al., *Oncogene*, 2012). Similarly, we observed that TRIM40-overexpressing cells displayed elongated and stretched morphology (A) or increased migration ability (B). (Please see the additional **Fig. #1** at the end of this response.) This is a good insight for further studies of the relationship between TRIM40 expression and cell polarity, and we incorporated additional statements into the text of the revised manuscript to reflect the reviewer's comments.

Fig. #1**A** (Supplementary Fig. 2b of the revised manuscript)**B**
* **Note to Figures:** (Left) DIC images showing that TRIM40-overexpressing cells displayed elongated and stretched membrane morphologies (red box and arrows in enlarged images). Scale bars, 100 μ m. (Right) Representative images and statistical results of Transwell migration assays of Mock or TRIM40-overexpressing HT-29 cells, *** P <0.001 (Student's t -test). Scale bars, 100 μ m. Immunoblots showing TRIM40 expression levels in cells used in Transwell migration assay.

• **Nuclear localization of TRIM40 should be discussed.**

As

the reviewer pointed out, the subcellular pattern of TRIM40 seems to be distributed in both the cytosol and the nucleus. Because the images in these datasets were orthogonal combinations of entire stacks of confocal images, we attempted to analyze each z-stack image separately to accurately discern the subcellular localization of TRIM40. We observed that TRIM40 was predominantly distributed in the cytoplasm, and it showed faint fluorescence within the nuclei in the equatorial plane that contained the center of the nucleus. (Please see the additional **Fig. #1** at the bottom of this response.) To clarify this result, we performed additional experiments to measure TRIM40 expression in the cytoplasmic and nucleoplasmic fractions. Indeed, TRIM40 was predominantly detected in the cytoplasm, and it was very faintly observed in the nucleoplasmic fraction by immunoblotting with a longer exposure time. (Please see the additional **Fig. #2** at the bottom.) We included additional statements in the text of the revised manuscript to reflect the reviewer's point.

Fig. #1**Fig. #2**
* **Note to Figures:** (Left) Representative z-stack confocal images of TRIM40 localization in TRIM40-expressing HT-29 cells. Scale bars, 100 μ m. (Right) Immunoblots showing TRIM40 expression levels in cytoplasmic (CA) or nucleoplasmic (NS) fractions in Myc-TRIM40-expressing HT-29 cells.

• **As mentioned in my first comment, expression of TRIM40 using an specific antibody would be informative. This also applied to the expression in Control-HT29, and teh validation of the main findings with an specific anti-TRIM40 antibody.**

As mentioned above, we incorporated new IHC results with an anti-TRIM40 antibody into the **new Fig. 1f** of the revised manuscript. In addition, as the reviewer suggested, we performed immunoblot analysis with an anti-TRIM40 antibody in control and TRIM40-expressing HT-29 cells. As expected, because there was no detectable TRIM40 expression in the control HT-29 cells, which is consistent with our previous datasets and the results of *Noguchi et al. (Carcinogenesis, 2011)*, we could not conduct further *in vitro* experiments with HT-29 cell lines by using an anti-TRIM40 antibody. (Please see the additional **Fig. #1** at the right of this response; red box and arrow)

• **CytoD induced a different pattern than TRIM40 overexpression and/or PMA, as indicated in ED Dig. 3d. I'm not so sure about cortical actin retraction (Fog.2a), and this is simply a matter of different cell shape. This should be discussed in the text.**

It is well known that cortical actin appears to be disrupted in Cyto.D-treated epithelial cells, which display a reduction of cofilin-1 phosphorylation and subsequent F-actin depolymerization (Faux et al., *Journal of Cell Science*, 2004; Randen L. Patterson et al., *Cell*, 1999; Joana N. Bugalhao, *MicrobiologyOpen*, 2016, Evelyn Garlick et al., *Scientific Reports*, 2022, Minjung Kim et al., *Experimental and Molecular Medicine*, 2012). While investigating the pathological role of TRIM40 in IBD, we repeatedly observed that TRIM40-overexpressing HT-29 cells exhibited irregular F-actin accumulation around the nuclear membrane and cortical actin disruption, which seemed to be quite similar to Cyto.D-treated and PMA-treated cells (Amitava Mukherjee et al., *The journal of biological chemistry*, 2009; Qing Yang et al., *Molecular Biology of the Cell*, 2013; please see the additional **Fig. #1** at the end of this response; red boxes) These results together with the RNA-seq results showing the correlation with the cytoskeleton led us to speculate on a possible role of TRIM40 in aberrantly altering cortical actin polymerization as follows: "These findings indicate that TRIM40 may be directly involved in regulating actin polymerization". Indeed, we observed that TRIM40 directly targets ROCK1 for degradation and facilitates CFL1 dephosphorylation and F-actin depolymerization, which is similar to the influence of Cyto.D in CFL-1 dephosphorylation-mediated F-actin disruption.

To avoid any confusion, we provided more convincing and sharper images clearly showing irregular F-actin accumulation around the nuclear membrane and cortical actin disruption in Cyto.D-treated cells (**new Supplementary Fig. 3d**).

* **Note to Figure #1.** Fig. 5 from the published data [Faux et al., *Journal of Cell Science*, 2004]

Human epithelial colon cancer cells (SW480) treated with Cyto.D exhibited aberrant accumulation of F-actin around the nuclear membrane and disruption of cortical actin formation [**red box** (Cyto.D-treated), compared with **blue box** (control SW480)]. Vehicle control-treated SW480 (A) and Cyto.D-treated SW480 (G) were co-stained with Rhodamine-Phalloidin to visualize F-actin (A and G). *APC; the adenomatous polyposis col tumor suppressor gene.

• ***I think the data concerning TEER lack relevance, since there is a clear issue concerning cell adherence. do not form a monolayer; it would then be important to show that TRIM40 overexpressing cells used for TEER measurement reach confluence. Otherwise, the TERER data should not be shown.***

The reviewer raised a concern that TRIM40 overexpression might affect cell growth reaching confluence in comparison with control cells. Although we carried out TEER assays upon reaching 100% confluence with both control and TRIM40-overexpressing cells, we agree with the reviewer's concern. To clarify this point, we established a Tet-On system for doxycycline-inducible TRIM40 expression, which allowed us to transiently overexpress TRIM40 and accurately monitor the specific effect of TRIM40 overexpression on epithelial monolayer confluence. Consistently, we observed that the epithelial membrane integrity was disrupted in cells that were maximally overexpressing TRIM40 after doxycycline treatment, as indicated by measurements showing that electrical resistance was inhibited by over 80% in the epithelial cells. Therefore, we are now confident that TRIM40 disrupts epithelial integrity. We have included these new results in the **new Fig. 2f** and **new Supplementary Fig. 4d** of the revised manuscript.

• ***Relevance of LPS injected mouse model; if the authors considered that this is a relevant observation, the n number should be increased. What is the added value of these data?***

We agree with the reviewer's comment. To provide more convincing results, we repeated this experiment with an increased number of mice. We replaced the previous results with the new datasets in the **new Supplementary Fig. 7d** of the revised manuscript.

• ***I definitely think that this study would benefit from data generated with TRIM40KO mice***

subjected to other colitis model where epithelial integrity is important.

As the reviewer suggested, we attempted to use a 2, 4, 6-trinitrobenzene sulfonic acid (TNBS)-induced colitis system, which is used to develop mucosal inflammation driving colitis. Although C57BL/6 (B6) mouse strain tends to be more resistant to TNBS-induced colitis than BALB/c (Philip Alex et al., *Inflammatory Bowel Diseases*, 2008; Matthias A. Engel et al., *Gastroenterology*, 2011; Janelle A Jiminez et al., *Gut Pathogens*, 2015; Stefan Wirtz, *Nature protocols*, 2017), following TNBS administration, wild-type B6 mice developed severe symptoms of IBD, such as loss of body weight, but these symptoms were not seen in *Trim40*-deficient B6 mice, which was consistent with the results with DSS administration. We also observed that TNBS-induced acute gene expression of the pro-inflammatory chemokine *Cxcl1* was substantially attenuated in *Trim40*-deficient mice. This is consistent with the results from the DSS-induced colitis model, suggesting that *Trim40*-deficient mice are resistant to both DSS-induced and TNBS-induced colitis. We included the additional TNBS *in vivo* datasets in the **new Supplementary Fig. 8e and g** of the revised manuscript.

• Some publications described ROCK expression upregulation in IBD; how this can be explained based on the present data; please, discuss.

We thank the reviewer for giving us valuable comments. As the reviewer pointed out, several studies have focused on the pathological role of ROCK1 in endothelial cells and intestinal M1 macrophages (Alexander Garcia Ponce et al., *Scientific Reports*, 2016; JEAN-PIERRE SEGAIN et al., *Gastroenterology*, 2003; Xuewen Wang et al., *Biochemical Pharmacology*, 2022). However, ROCK1 has been reported to be expressed in fibroblasts and epithelial, endothelial, and muscle cells within intestinal tissues (Tom Holvoet et al., *Gastroenterology*, 2017), suggesting that it may play a different role in each cell type. Indeed, our results further suggest that TRIM40 overexpression-mediated ROCK1 degradation facilitates F-actin depolymerization and intestinal integrity disruption, which occur during chronic inflammation-induced IBD pathogenesis.

In addition, it has been reported that ROCK2 inhibits M1 macrophage polarization, which is opposite to the effect of ROCK1 signaling, and functions as a negative regulator of the immune response by suppressing IRF3 activity (Souska Zandi et al., *Cell Reports*, 2015; Chi Zhang et al., *Science advances*, 2021). This is a good insight for further studies of the relationship between TRIM40 overexpression and ROCK2 expression/activity in IBD pathogenesis. Therefore, we included additional statements in the Discussion section of the revised manuscript to reflect the reviewer's points.

• Abstract: revise the sentence about the different mutants; was it only Δ RING-mut the one leading to recovery of ROCK1 and actin cytoskeleton; as it reads it seems that all three mutants affected ROCK1 degradation and actin cytoskeleton.

As the reviewer has highlighted, we showed that TRIM40 Δ RING fails to induce ROCK1 degradation, resulting in the partial restoration of ROCK1 protein levels and of cortical actin formation (Fig. 4c and e). However, in the original manuscript, we also showed that TRIM40 Δ BB

fully lost its binding affinity for ROCK1, leading to the loss of both TRIM40-driven ROCK1 degradation and cortical actin disruption (Fig. 4c and e). Additionally, we showed that TRIM40 Δ CT had no effect on ROCK1 degradation or interaction and eventually lost its cortical actin disruption functionality (Fig. 4c, e, and f). Moreover, neither TRIM40 Δ BB nor TRIM40 Δ CT exhibited negative effects on cell-cell distance, and the electrical resistance of cells was only partially affected by TRIM40 Δ RING (Fig. 4g and h), indicating that the functional abilities of the TRIM40 mutants to affect ROCK1 degradation were directly reflected in both cortical actin formation and cell-cell contact. Therefore, we had concluded that "A mutant TRIM40 lacking the RING finger, the B-box, or the C-terminal domain has an impaired ability to accelerate ROCK1 degradation-driven actin depolymerization" as stated in the abstract of the original manuscript.

Minor

- ***L81: its expression is in IBD samples...***

We thank the reviewer for identifying this error. We corrected the error in the revised manuscript.

- ***Figure 1 should be organized in a way that the subfigures follow an order; otherwise, it is complicated to read the figure***

We agree with the reviewer's comment. We carefully reorganized the subfigures in Fig. 1 of the revised manuscript.

- ***L88-89: the sentence "this notable upregulation in the expression of the otherwise epigenetically silenced TRIM40 occurred at a high frequency across a significant distribution of patients with IBD. "; is this based on the data shown in Fig 1.d. I think this is overstated, and I would consider rephrasing it to show that in this a rather small cohort you can see an upregulation of the expression.***

We agree with the reviewer's comment and have carefully rephrased this sentence in the revised manuscript.

- ***ED Fig. 2b: pictures from Caco2 are not visible; I would suggest choosing other representative pictures.***

We agree with the reviewer's comment and have replaced the Caco-2 images with more convincing and shaper images.

- ***L252-253: what does it mean DSS alone of DSS in conjunction with inflammation???***

We carefully rephrased this sentence in the revised manuscript to avoid confusion.

- ***L268: epithelial barrier***

We thank the reviewer for catching this error and have corrected the error in the revised manuscript.

- ***ED 7d: picture from distal colon, WT DSS, is it from the same magnification as the other pictures???***

We thank the reviewer for catching this error. We replaced these images with images that all have the same magnification. We also carefully double-checked all confocal images throughout the revised manuscript.

- ***L300-301: DSS accompanying inflammation itself is not an essential process in IBD: what does this sentence mean?***

We agree with the reviewer's point about ambiguity. To avoid confusion, we carefully amended this sentence in the revised manuscript.

REVIEWERS' COMMENTS

Reviewer #1 (Remarks to the Author):

The updated version of this manuscript has fully addressed my questions. The data is solid and supports their conclusions.

Reviewer #2 (Remarks to the Author):

The authors have very nicely addressed key concerns raised in the previous review.

Reviewer #3 (Remarks to the Author):

The authors have successfully addressed my comments; I recommend the current version for publication

Because we have successfully addressed all remaining concerns and there have been no further comments, we only uploaded the revised manuscript to address the editorial requests.

REVIEWERS' COMMENTS

Reviewer #1 (Remarks to the Author):

The updated version of this manuscript has fully addressed my questions. The data is solid and supports their conclusions.

Reviewer #2 (Remarks to the Author):

The authors have very nicely addressed key concerns raised in the previous review.

Reviewer #3 (Remarks to the Author):

The authors have successfully addressed my comments; I recommend the current version for publication.